# Administrative-driven hierarchical management of atrial fibrillation on cardiovascular events: a prospective matched cohort study

Mu Chen [1], Mingzhe Zhao[1], Yuli Yang[1], Xin Cui[2], Chunfang Wang[3], Tian Xia[3], Wenqi Tian[2], Peng Liao[4], Yudong Fei[1], Peng-Cheng Yao[1], Xiaoxiao Zhu[5], Yongbo Wu[5], Mei Yang[1], Jian Sun[1], Li Luo[4], Hong Wu[6], Qunshan Wang[1] ✉ & Yi-Gang Li [1] ✉

The administrative-driven hierarchical management (ADHM) refers to the management model that general practitioners act as "voluntary gatekeepers" under local government policy incentives in the non-mandatory hierarchical medical system. However, the impact of ADHM of atrial fibrillation (AF) on cardiovascular events remains unclear. We enrolled 1455 patients with AF in the ADHM cohort and 7275 in the matched usual care (UC) cohort in Shanghai, China. During the 30 months follow-up, the rate of the primary outcome (the composite of ischemic stroke, systemic embolism, myocardial infarction, major bleeding, acute heart failure and cardiovascular death) was significantly lower in the ADHM cohort than the UC cohort (hazard ratio: 0.624; 95% confidence interval=0.554-0.703; p < 0.0001). Comparisons of the secondary endpoints, including all-cause death, cardiovascular death, ischemic stroke, and acute heart failure, also favored the ADHM cohort, while non-cardiovascular death, systemic embolism, myocardial infarction, and major bleeding were similar between cohorts. The medical costs per patient per survival day were lower in the ADHM cohort. In sum, the ADHM model is effective in reducing cardiovascular events in patients with AF. Trial number: ChiCTR2000036931

Atrial fibrillation (AF) is the most common arrhythmia worldwide, characterized by recurrent episodes that not only cause discomfort but also lead to complications such as stroke and heart failure (HF). These complications result in increased morbidity and mortality, imposing a heavy burden on society[1–3]. Nowadays, AF is increasingly recognized as a "chronic syndrome" that requires long-term

comprehensive management. In Europe and the Asia-Pacific region, the Atrial Fibrillation Better Care (ABC) pathway has therefore been proposed, which is implemented with 3 components: "A"- Anticoagulation/Avoid stroke; "B"- Better symptom control; and "C"- Cardiovascular risk factor and comorbidities management[4,5]. The European guideline also highlight two steps of confirmation and

[1]Department of Cardiology, Xinhua Hospital, School of Medicine, Shanghai Jiao Tong University, Shanghai, China. [2]Shanghai Health Statistics Center, Shanghai, China. [3]Shanghai Municipal Center for Disease Control & Prevention, Shanghai, China. [4]School of Public Health, Fudan University, Shanghai, China. [5]Shanghai Siwei Medical Co. Ltd., Shanghai, China. [6]Shanghai Municipal Health Commission, Shanghai, China. ✉ e-mail: wangqunshan@xinhuamed.com.cn; liyigang@xinhuamed.com.cn

characterization (CC) of AF preceding ABC pathway, forming a complete model for integrated AF care, "CC-to-ABC"[4]. The implementation of the CC-to-ABC pathway has been found to be associated with a reduction in cardiovascular events and mortality, highlighting the significance of this strategy[6–8]. However, in real-world practice, the sustainable adherence to the CC-to-ABC pathway was poor, leading to high risks of cardiovascular events and death among non-adherent patients[6,7,9,10].

Integrating AF management into the hierarchical healthcare system might be an effective approach to promote the implementation of the CC-to-ABC pathway. To optimize the rational allocation of healthcare resources, the National Health Commission of China has actively initiated a hierarchical diagnosis and treatment model, encouraging the coordinated patient management across healthcare institutions at different levels. However, unlike the systems in the United Kingdom and United States, China's hierarchical healthcare system is not mandatory, without the support of a designated family doctor or commercial health insurance-driven cost-saving considerations from both sides of patients and insurance companies. In China, although the reimbursement rates at the community healthcare centers (CHCs) are higher than those at higher-level hospitals (by approximately 10%) for patients, patients especially those with cardiovascular diseases show stronger preference seeking healthcare services at large academic hospitals[11]. As general practitioners (GPs) at CHCs do not play a mandatory gatekeeping role, patients could freely choose to visit higher-level hospitals even for first consultations without the need of referrals from GPs at non-critical situations. This results in significant patient congestion at higher-level hospitals, with large numbers of AF patients with mild symptoms crowding these institutions, consuming unnecessary resources meant for patients with critical conditions or requiring interventional therapies. In fact, patients with mild conditions also face challenges when receiving healthcare in large hospitals, including lack of familiarity with their conditions by the specialists, limited time for consultation and health education, and potential conflicts with previous treatment plans, which can hinder the continuity, consistency, and adherence of treatment. Therefore, a practical scheme suitable for CC-to-ABC implementation in low-resource regions such as China is urgently needed in the absence of mandatory hierarchical medical systems.

To improve the local application of hierarchical medical systems, the Shanghai Municipal Health Commission (SHMHC), in collaboration with medical and public health experts, proposed various hierarchical management models for various chronic diseases. The AF management model, i.e., the administrative-driven hierarchical management (ADHM) of AF, was largely adapted from the CC-to-ABC pathway and also referred as the "ACC-to-ABC" pathway. The added "A" prior to the "CC-to-ABC" has threefold meanings, i.e., Administration, Association, and AF Center Union. Specifically, "Administration" refers to government-issued administrative directives and policy support. Based on the management situation of AF in Shanghai by recent epidemiological survey[2], the SHMHC issued the *White Paper on the Standardized Prevention and Treatment of Atrial Fibrillation in Shanghai*, which authoritatively determined the standard operating procedure for the hierarchical management scheme for AF in Shanghai. This was facilitated by a series of supporting policies and regulations towards CHCs and GPs, including drug supply and extended prescription, regular performance evaluation, establishment of an incentives and punishment system, and development and authorization of an electronic "Shanghai AF Management Platform", etc. "Association" refers to the academic support from medical associations including the Shanghai Medical Association, the Shanghai Physician Association, and Shanghai Stroke Society, etc. These associations supported the design of the Platform, knowledgebase updating, GP training, and patient education. "AF Center Union" consisted of secondary and tertiary hospitals which equipped with electrophysiological laboratories

and cardiac intensive care units. These hospitals established medical alliance with CHCs in the neighborhood regions, ensuring a smooth bidirectional referral that patients being referred from CHCs for interventional procedures or complication management and being referred back to CHCs post-procedure or when their conditions were stabilized. Specialists from the AF Center Union also assisted GPs in clinical decision making as required. Thus, in the ADHM model (the ACC-to-ABC pathway), CHCs serve as the primary venues and GPs from CHCs as the main force for AF management, supported by the local health administration, academic associations, and higher-level hospitals (Supplementary Fig. 1).

Given that it is difficult to reform the national healthcare system in short term, administrative intervention by the local government might be a feasible option to implement the hierarchical system under the non-mandatory settings, especially in low-and-middle-income countries (LMICs) where public healthcare as the mainstay, such as China, India, Brazil, and Kenya, etc[12–14]. In this work, we show, the ADHM model (or ACC-to-ABC pathway) for AF is effective in reducing cardiovascular diseases and mortality. With policy incentives by the local government, the GPs acting as "voluntary gatekeepers" in the hierarchical system provide high-quality healthcare service to patient with AF, thereby reducing the burden on large hospitals and medical costs. This effective, adoptable, and cost-saving strategy for AF management should be scaled up in other low-resource regions lacking mandatory hierarchical medical systems.

## Results

### Patient characteristics

Between August 1 and November 28, 2022, a total of 8730 patients with AF were included in the study, including 1455 in the ADHM cohort and 7275 in the matched usual care (UC) cohort. Patients in the ADHM cohort were enrolled from 34 CHCs in the pilot districts, including 543 patients from 12 CHCs in Xuhui District, 531 from 12 CHCs in Yangpu District, and 381 from 10 CHCs in Changning District. Patients in the UC cohort were residents of the other 13 districts of Shanghai (Fig. 1). Study participants were aged 76.0 ± 9.9 years and 45.9% female, with CHA2DS2-VASc score 4.96 ± 1.93 and HAS-BLED score 2.66 ± 0.98. Baseline characteristics were well balanced between groups (Table 1).

### Healthcare preferences

Within 12 months prior to the enrollment, the times of patients visiting CHCs were similar between groups. After the enrollment, the proportion of patients with regular visiting to CHCs (≥4 times) were significantly higher in the ADHM cohort than the UC cohort (77.5% vs. 50.3%, p < 0.001). Correspondingly, the times of patients visiting higher-level hospitals were similar between groups before enrollment (3.98 ± 2.63 vs 3.87 ± 2.67, p = 0.164), but significantly reduced in the ADHM cohort (1.96 ± 1.67 vs. 3.62 ± 2.49, p < 0.001, Fig. 2).

### Treatment

The proportions of patients received at least one anticoagulant prescription were similar between groups before enrollment (58.4% vs. 58.1%, p = 0.801), which became significantly higher in the ADHM cohort than the UC cohort during follow-up (89.5% vs. 60.9%, p < 0.001). Likewise, regular anticoagulation rates were similar before enrollment (40.5% vs. 40.4%, p = 0.930) and increased drastically in the ADHM cohort after enrollment, compared with the UC cohort (78.6% vs. 46.3%, p < 0.001). In addition, more patients received left atrial appendage closure (LAAC) procedure, the alternative strategy of stroke prevention, in the ADHM cohort (3.6% vs. 1.4%, p < 0.001). As for the symptom control process, more patients were on antiarrhythmic drugs (43.8% vs. 37.7%, p < 0.001) and catheter ablation (12.9% vs. 4.0%, p < 0.001) in the ADHM cohort than the UC cohort, respectively (Fig. 3).

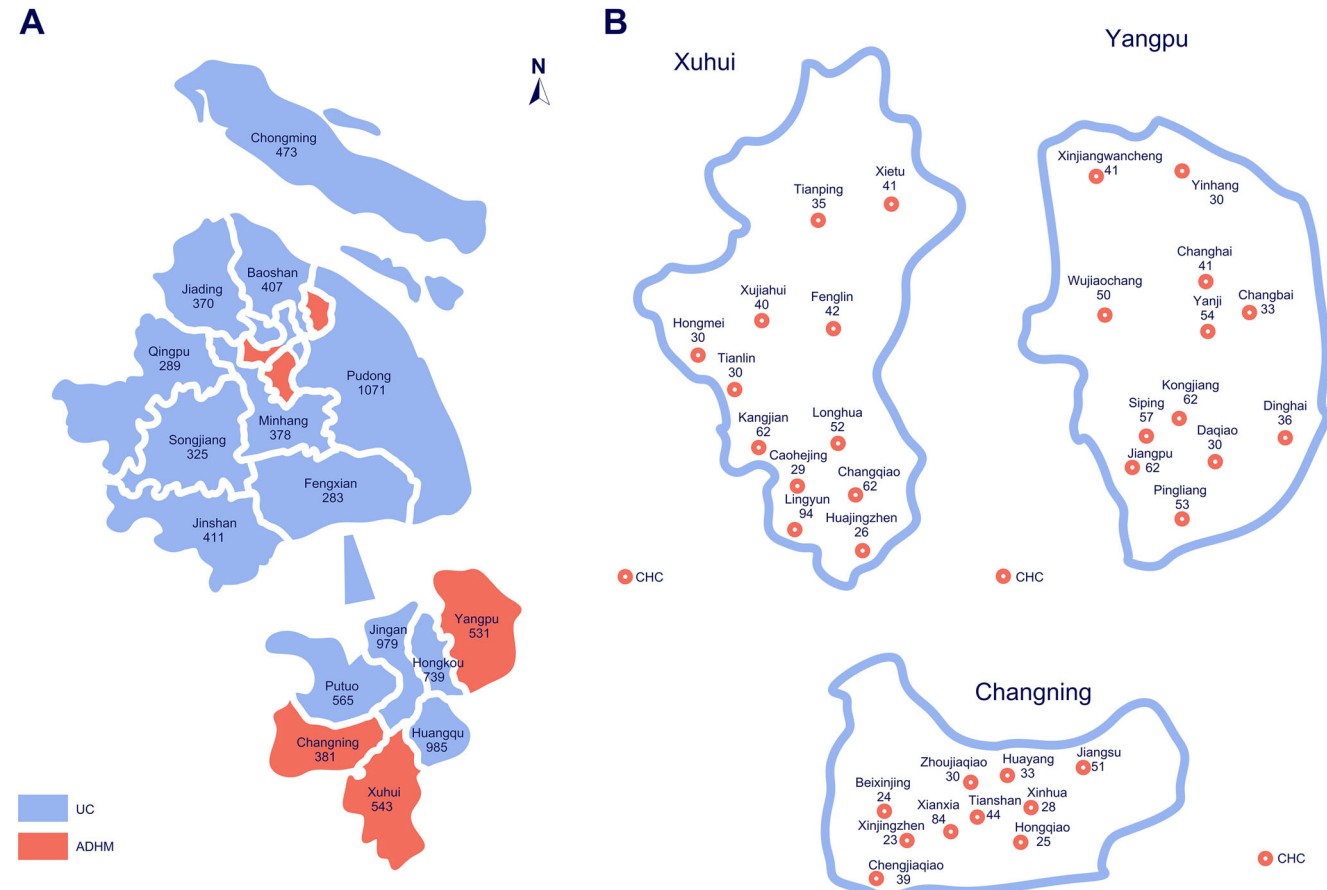

**Fig. 1 | Geographic locations of study populations. A** Maps showed districts implementing ACC-to-ABC pathway (ADHM cohort, red) and usual care (UC cohort, blue). **B** Maps showed CHCs (red circle) implementing ACC-to-ABC pathway in Xuhui, Yangpu, and Changning Districts. Numbers indicate the number of patients in each district and CHC enrolled in the study. Source data are provided as a Source Data file.

## Outcomes

During the 30 months follow-up period, the primary outcome, i.e., the composite of ischemic stroke, systemic embolism (SE), myocardial infarction (MI), major bleeding, acute heart failure (HF), and cardiovascular (CV) death, was confirmed in 219 participants (6.02% rate per year) in the ADHM cohort and 1664 subjects (9.15% rate per year) in the UC cohort (HR with intervention: 0.624; 95% CI = 0.554–0.703; $p < 0.0001$; Fig. 4). The reduced risk of the primary outcome was observed both before/during and after the shift in China's COVID-19 policies (Supplementary Fig. 2). Compared with the UC cohort, the ADHM cohort also showed reduced risks in the prespecified secondary outcomes, including the composite of ischemic stroke, SE, MI, major bleeding, acute HF, and all-cause death (HR = 0.674; 95% CI = 0.604-0.752; $p < 0.0001$), all-cause death (HR = 0.683; 95% CI = 0.592-0.789; $p < 0.0001$), CV death (HR = 0.611; 95% CI = 0.510-0.732; $p < 0.0001$), ischemic stroke (HR = 0.583; 95% CI = 0.475-0.717; $p < 0.0001$), and acute HF (HR = 0.702; 95% CI = 0.531-0.928; $p = 0.0027$). The risks of non-CV death, SE, MI, and major bleeding were similar between the ADHM and UC cohorts (Supplementary Fig. 3). Multivariate analysis showed under UC management, age, history of stroke/transient ischemic attack (TIA), HF, periphery artery disease, and the times of visit to CHCs were predictor of the primary outcome in the entire study population (Supplementary Table 1). For the ADHM cohort, age, history of major bleeding, and liver deficiency were risk factors, while regular anticoagulant prescription was the protective factor of the primary outcome (Supplementary Table 2). Age, history of stroke/TIA, periphery artery disease, and LAAC were predictors for the primary outcome in the UC cohort (Supplementary Table 3).

A total of 161 deaths (4.43% rate per year), including 93 CV deaths and 68 non-CV deaths, occurred among 1455 participants in the ADHM cohort, while 1158 deaths (6.37% rate per year, 742 CV deaths and 416 non-CV deaths) among 7275 patients in the UC cohort during follow-up. The rates of all-cause death and CV death were lower in the ADHM cohort than the UC cohort (11.06% vs.15.92%, $p < 0.001$; 6.39% vs. 10.20%, $p < 0.001$; respectively), while the rates of non-CV death were similar between cohorts (4.67% vs. 5.72%, $p = 0.112$). The reduced CV deaths in the ADHM cohort were primarily driven by the reductions in deaths caused by ischemic heart disease (3.90% vs. 5.20%, $p = 0.041$) and ischemic stroke (1.58% vs. 3.81%, $p < 0.001$, Fig. 5).

## Medial costs

Compared with the UC cohort, the medical costs per patient per survival day were lower in the ADHM cohort (213 ± 266 vs. 263 ± 638 Chinese Yuan, $p < 0.001$, Table 2), which was attributed to markedly reduced costs in higher-level hospitals (156 ± 191 vs. 211 ± 575 Chinese Yuan, $p < 0.001$). The costs in outpatient were higher (60 ± 69 vs. 45 ± 98 Chinese Yuan, $p < 0.001$) while the costs in inpatient or emergency department were drastically reduced (153 ± 255 vs. 218 ± 623 Chinese Yuan, $p < 0.001$) in the ADHM cohort than the UC cohort. The total cost during the entire follow-up was shown in the Supplementary Table 4.

## Discussion

This study has shown that the implementation of ADHM model (ACC-to-ABC pathway) reduces risks of cardiovascular events in patients

**Table 1 | Baseline Characteristics**

| | Total (n = 8730) | ADHM cohort (n = 1455) | UC cohort (n = 7275) | P value |
|---|---|---|---|---|
| Female sex (n, %) | 4008 (45.9) | 668 (45.9) | 3340 (45.9) | 1.000 |
| Age (years) | 76.0 ± 9.9 | 76.0 ± 9.5 | 76.0 ± 9.9 | 0.848 |
| CHA2DS2-VASc | 4.96 ± 1.93 | 4.96 ± 1.96 | 4.96 ± 1.93 | 0.970 |
| HAS-BLED | 2.66 ± 0.98 | 2.67 ± 0.99 | 2.66 ± 0.98 | 0.201 |
| Hypertension (n, %) | 8103 (92.8) | 1339 (92.0) | 6764 (93.0) | 0.201 |
| Diabetes (n, %) | 4758 (54.5) | 795 (54.6) | 3963 (54.5) | 0.908 |
| Stroke/TIA (n, %) | 4476 (51.3) | 748 (51.4) | 3728 (51.2) | 0.590 |
| Systemic embolism (n, %) | 300 (3.4) | 57 (3.9) | 243 (3.3) | 0.270 |
| Heart failure (n, %) | 4584 (52.5) | 757 (52.0) | 3827 (52.6) | 0.687 |
| Peripheral artery disease (n, %) | 307 (3.5) | 62 (4.3) | 245 (3.4) | 0.091 |
| Myocardial infarction (n, %) | 268 (3.1) | 49 (3.4) | 219 (3.0) | 0.471 |
| Valvular heart disease (n, %) | 288 (3.3) | 51 (3.5) | 237 (3.3) | 0.630 |
| COPD (n, %) | 1940 (22.2) | 327 (22.5) | 1613 (22.2) | 0.800 |
| OSAHS (n, %) | 69 (0.8) | 16 (1.1) | 53 (0.7) | 0.144 |
| History of major bleeding (n, %) | 971 (11.1) | 173 (11.9) | 798 (11.0) | 0.308 |
| History of cancer (n, %) | 1083 (12.4) | 179 (12.3) | 904 (12.4) | 0.896 |
| Renal deficiency (n, %) | 444 (5.1) | 83 (5.7) | 361 (5.0) | 0.239 |
| Liver deficiency (n, %) | 1404 (16.1) | 231 (15.9) | 1173 (16.1) | 0.815 |
| Medication history | | | | |
| Regular anticoagulation* | 3531 (40.4) | 590 (40.5) | 2941 (40.4) | 0.930 |
| Warfarin (n, %) | 941 (10.8) | 156 (10.7) | 785 (10.8) | 0.938 |
| Dabigatran (n, %) | 902 (10.3) | 160 (11.0) | 742 (10.2) | 0.362 |
| Xa factor inhibitor (n, %) | 2250 (25.8) | 394 (27.1) | 1856 (25.5) | 0.212 |
| Antiplatelet drug (n, %) | 2935 (33.6) | 472 (32.4) | 2463 (33.9) | 0.297 |
| Antiarrhythmic drugs (n, %) | 2876 (32.9) | 480 (33.0) | 2396 (32.9) | 0.968 |
| Rate control drugs (n, %) | 3320 (38.0) | 576 (39.6) | 2744 (37.7) | 0.180 |
| Statin (n, %) | 2429 (27.8) | 397 (27.3) | 2032 (27.9) | 0.616 |
| ACEi/ARB/ARNi (n, %) | 3902 (44.7) | 650 (44.7) | 3252 (44.7) | 0.985 |
| MRA (n, %) | 968 (11.1) | 167 (11.5) | 801 (11.0) | 0.604 |
| SGLT2i (n, %) | 671 (7.7) | 108 (7.4) | 563 (7.7) | 0.679 |
| History of cardiac procedures | | | | |
| Catheter ablation (n, %) | 957 (11.0) | 162 (11.1) | 795 (10.9) | 0.818 |
| LAAC (n, %) | 107 (1.2) | 18 (1.2) | 89 (1.2) | 0.965 |
| Cardioversion (n, %) | 73 (0.8) | 14 (1.0) | 59 (0.8) | 0.563 |

*ACEi* angiotensin-converting enzyme inhibitor, *ADHM* administrative-driven hierarchical management, *ARB* angiotensin II receptor blocker, *ARNi* angiotensin receptor-neprilysin inhibitor, *CHA2DS2-VASc* congestive heart failure, hypertension, age 75 [doubled], diabetes mellitus, prior stroke or transient ischemic attack [doubled], vascular disease, age 65–74, female, *COPD* chronic obstructive pulmonary disease; *HAS-BLED* hypertension, abnormal renal and/or liver function, previous stroke, bleeding history or predisposition, labile international normalized ratios, elderly, and concomitant drugs and/or alcohol excess; *LAAC*= left atrial appendage closure; *MRA*= mineralocorticoid receptor antagonist, *OSAHS* obstructive sleep apnea-hypopnea syndrome, *SGLT2i* sodium-glucose co-transporter 2 inhibitor, *TIA* transient ischemic attack, *UC* usual care. * Regular anticoagulation indicates patients had at least three prescriptions of oral anticoagulants within 12 months before enrollment. The independent Student's *t*-tests and the χ2 tests were adopted for continuous and categorical variables, respectively. Two-sided *P* value was adopted. Source data are provided as a Source Data file.

with AF. The net group difference in regular anticoagulation rate increase of 32.2%, along with rates of antiarrhythmic prescriptions increase of 8.5% and ablation increase of 8.7%, was associated with a 37% (95% CI 30–45) reduction in the hazard of the composite of major cardiovascular events. The ADHM model was also associated with 32% (21–41) reduction in the hazard of all-cause death, 39% (27–49) of CV death, 42% (28–53) of ischemic stroke, and 30% (7–47) of acute HF, respectively. In addition, under the ADHM model, patients with AF visited primary healthcare facilities more regularly, thereby reducing the burden on high-level hospitals. The medical costs per patient per survival day were also reduced by nearly 50 Chinese Yuan under ADHM model.

As the population ages globally, the prevalence of AF is increasing, not only in Western societies but also in LMICs, including China[1–3]. Our previous investigation revealed that the prevalence of AF in Shanghai among individuals aged over 65 was 2.4%, and 6.7% among those over

80, often coexisting with other chronic conditions such as hypertension, diabetes, and COPD, and closely associated with stroke, HF, and mortality[2]. In this study, we found that more than 58% of AF patients received at least one anticoagulant prescription in the year prior to enrollment; however, only about 40% were on regular anticoagulation. This indicated that even in Shanghai, a relatively developed city in China, the healthcare service for AF was fragmented, lacking continuity, which not only failed to prevent AF-related complications but also led to the inefficient use of healthcare resources.

In this study, we proposed a feasible management model for AF, which was driven by administrative intervention of local government, integrating resources to enable GPs to act as voluntary gatekeepers of the hierarchical medical system. The measures included: (1) releasing a white paper to clarify the centrality of GPs in the hierarchical management of AF and developing a standardized workflow; (2) incentivizing GPs to proactively manage and follow up patients with AF

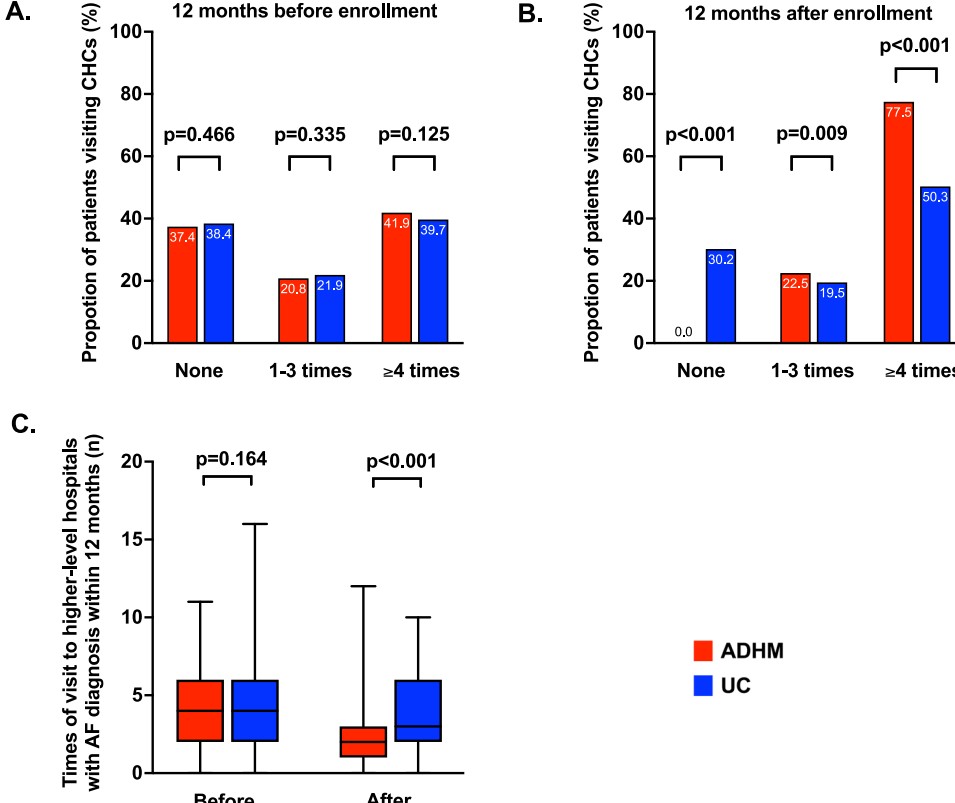

**Fig. 2 | Patients visiting CHC and higher-level hospitals.** *ADHM* administrative-driven hierarchical management, *AF* atrial fibrillation, *CHC* community healthcare center, *UC* usual care. For Fig. **c**, box plots show the median (center line), 25th and 75th percentiles (box bounds). The whiskers extend to the most extreme data points. The χ2 tests were applied for Fig. **A** and **B**, and independent Student's *t*-tests were adopted for Fig. **C**. Multiple comparisons were not made, and a two-sided *P* value was adopted. Source data are provided as a Source Data file.

through administrative directives, performance incentives, and reward-punishment systems; (3) designing an electronic platform to assist AF management, with three interfaces providing functionalities for government oversight, GP management, and patient education, respectively; (4) strengthening the regional healthcare network to facilitate referrals; (5) organizing training and education programs for GPs and patients, etc. Under the ADHM model, the healthcare-seeking behavior of patients with AF shifted significantly, with fewer visits to large academic hospitals and more regular visits to primary facilities. The reduction in higher-level hospital visits did not lead to a decrease in rates of anticoagulant and antiarrhythmic prescriptions but rather improved their regular use, indicating the consistency of treatment improved via the proactive and long-term management by GPs. In the ADHM cohort, fewer visits to hospitals, in turn, increased the volume of cardiac interventions, including ablation and LAAC performed in large hospitals, suggesting that selective referrals by GPs for patients with appropriate indications were more efficient than those made by patients themselves.

The ADHM model significantly reduced the incidence of ischemic stroke, acute HF, mortality, and their composite, without increasing bleeding risks. This was in consistent with the causes of death analysis that reduced death was caused by ischemic heart diseases and stroke in the ADHM cohort. Multivariate analysis showed the ADHM model was a protective factor of cardiovascular events independent of other factors such as age and comorbidities, while regular anticoagulant prescription was the driving factor for reduced cardiovascular outcomes in the ADHM cohort. Those results suggest the ADHM model provides a more consistent healthcare, especially standardized anticoagulation therapy, markedly improving the cardiovascular outcomes of patients with AF. In addition, due to the effective prevention of complications, improved efficiency of healthcare service, and optimized allocation of medical

resources, the ADHM model was cost-effective than UC by reducing the medical costs per patient per survival day, especially the costs of higher-level hospital and inpatient or emergent visits, which was in line with the prediction by the mathematical model[15].

These findings have important public health implications worldwide, especially in the city regions of the LMICs where public healthcare is the mainstay but lacks mandatory hierarchical systems, such as China, India, Brazil, Vietnam, the Philippines, and Kenya, etc[12–14]. In these countries, the implementation of hierarchical systems typically relies on government policies and planning; however, they are not mandatory. Therefore, in practical operation, patients have a certain degree of autonomy to select healthcare facilities, and often skip primary healthcare providers and go directly to specialists in large academic hospitals. This tendency not only results in wastage and undesirable encroachment of inherently scarce and unevenly allocated medical resources, but also affects the quality and continuity of the healthcare service. Given that it is difficult to reform the national healthcare system in a short period of time, administrative policy guidance by the local government to mobilize GPs as a voluntary gatekeeper, along with improving primary healthcare networks, enhancing medical service quality, and strengthening patient education, might become a feasible way to promote the implementation and sustainable development of hierarchical healthcare system through non-compulsory means. However, as Shanghai is a relatively economically developed city, whether the Shanghai-based ADHM intervention can generalize to rural China or other LMICs without government-driven healthcare needs further investigation.

## Limitations
First, the follow-up period coincided with the shift in China's COVID-19 policies from the "dynamic zero-COVID" to full reopening, resulting in

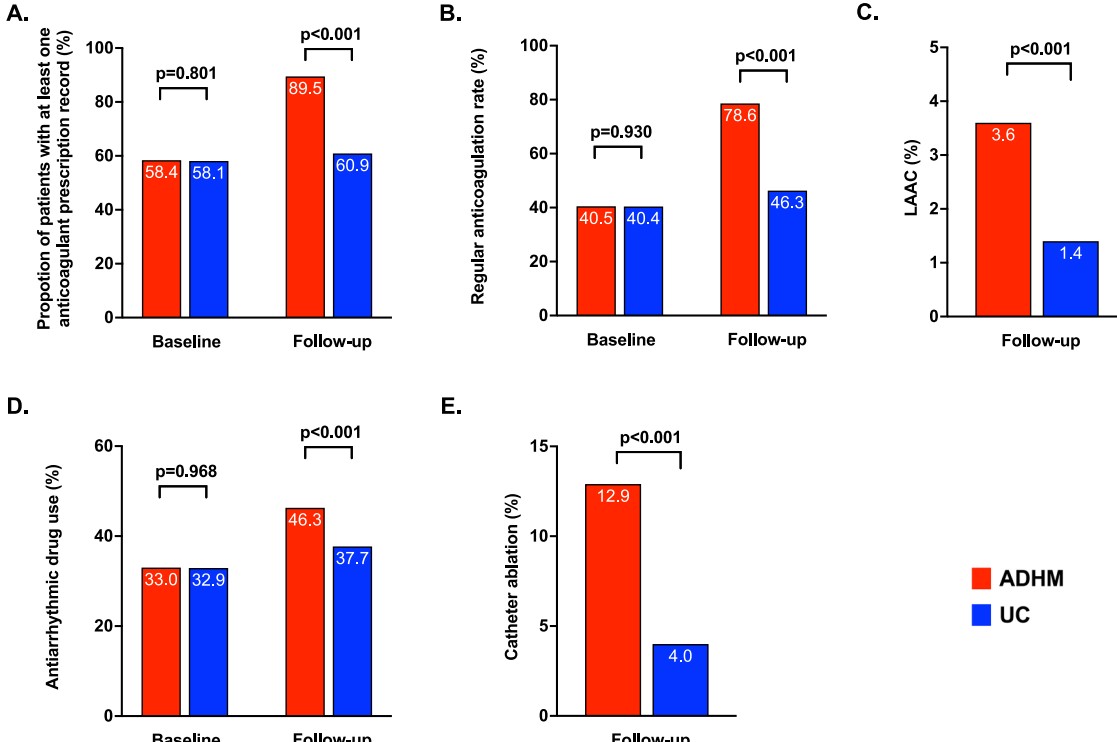

**Fig. 3 | Proportion of patients receiving stroke prevention and symptom control therapies.** Proportion of patients receiving at least one anticoagulant prescription (**A**), regular anticoagulation (**B**), LAAC (**C**), antiarrhythmic drugs (**D**), and catheter ablation (**E**) at baseline and during follow-up. ADHM administrative-driven hierarchical management, LAAC left atrial appendage closure, UC usual care. The $\chi^2$ tests were applied, multiple comparisons were not made, and two-sided *P* value was adopted. Source data are provided as a Source Data file.

a large wave of Omicron variant infections and excess mortalities within a focal period (approximately from November 2022 to January 2023). Consequently, a sharp drop in the survival curve was observed during the corresponding period. During the outbreak, patients' behaviors in healthcare seeking might be termed irrational, and GPs themselves were also widely infected, leading to an interruption in the ADHM implementation[16]. Additionally, anti-SARS-CoV-2 medications (such as nirmatrelvir/ritonavir) cannot be used in conjunction with rivaroxaban or amiodarone, and dosage adjustment for other anticoagulants was also required, leading to interruptions or deviations from standard therapies[17]. Hence, the impact of COVID-19 on this study is undeniable. However, as the Kaplan-Meier estimates of the primary outcome showed consistency before/during and after the Omicron outbreak, such an impact was reasonably assumed to be similar between both groups. Second, endpoint events were identified through diagnostic code scanning and medical record confirmation. The SHMHC database included medical records from all healthcare sources in the city, and of those with cross-provincial medical bill settlements, making this follow-up method highly reliable. However, this approach might still underreport events in patients who did not seek medical care or underwent overseas treatment. Nonetheless, the mortality should be fully accurate, as it was verified through cross-referencing by the health, public security, and civil affairs departments via the Shanghai Center for Disease Control and Prevention (CDC) database. Third, this study was not a randomized trial due to the administrative process involved. However, the two groups were propensity score-matched for sex, age, and comorbidities, and the treatment before enrollment was balanced between the groups. Fourth, the pattern of AF, i.e., paroxysmal, persistent, or permanent AF, could not able to be differentiated due to the adoption of International Classification of Diseases (ICD)-10 codes in the database rather than ICD-11 codes. The status of rhythm (AF and its burden or sinus rhythm) at the

end of the study, socioeconomic status, and other health behaviors were also not available in the database.

## Methods

### Ethical approval
This study was conducted in accordance with the Declaration of Helsinki. The study was approved by the institutional review board of Xinhua Hospital, School of Medicine, Shanghai Jiao Tong University, China, and registered with chictr.org.cn, Chinese Clinical Trial Registry (ChiCTR2000036931). The study was also authorized by the SHMHC through the administrative documents. Written informed consent was obtained from participants from the ADHM cohort before any study-related procedures.

### Patient enrollment
The study consisted of an intervention group (ADHM cohort) and a matched control group (UC cohort). The flowchart was shown in Fig. 6. Inclusion criteria were as follows: patients with a diagnosis of AF, aged over 18 years, and residents of Shanghai with government-issued medical insurance. Exclusion criteria included subjects with a life expectancy of less than three months, pregnant, or patients requiring immediate hospitalization or already receiving inpatient care. Detailed eligibility criteria are included in the Supplementary Table 5.

The ADHM cohort employed the hierarchical management model of AF driven by administrative interventions from the local government (ACC-to-ABC pathway) across all 34 CHCs at Xuhui, Yangpu, and Changning districts in Shanghai. Patients signed informed consent at their corresponding CHCs. Upon enrollment, GPs created a health profile for each patient on the electronic "Shanghai AF Management Platform", conducted risk stratification, initiated standardized treatment (including anticoagulation, symptom control, and management of AF-related comorbidities/complication/risk factors, as appropriate),

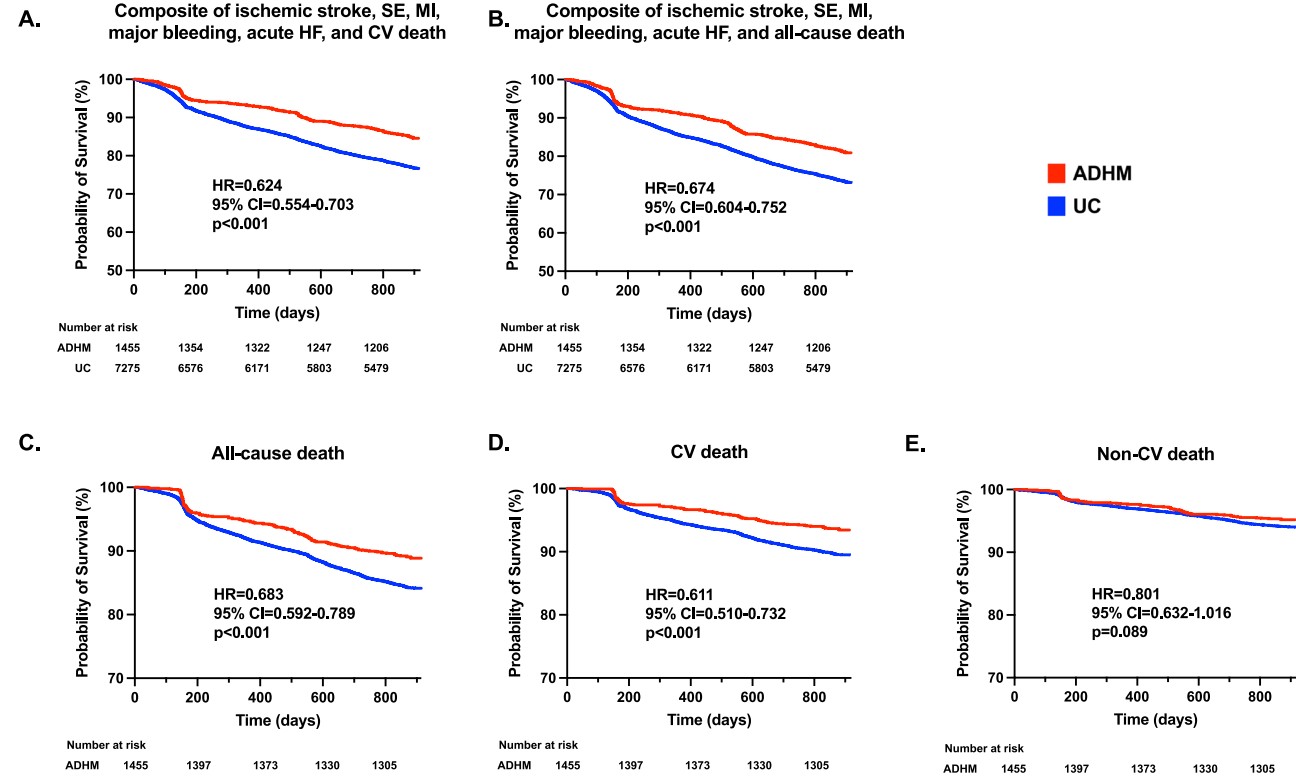

**Fig. 4 | Outcomes of patients with AF.** Shown are Kaplan-Meier estimates of the endpoint events, including the composite of CV death, ischemic stroke, SE, MI, major bleeding, and acute HF (primary endpoint, **A**), the composite of all-cause death, ischemic stroke, SE, MI, major bleeding, and acute HF (**B**), all-cause death (**C**), CV death (**D**), non-CV death (**E**), ischemic stroke (**F**), SE (**G**), MI (**H**), major bleeding (**I**), and acute HF (**J**) in the ADHM cohort (red curve) and the UC cohort (blue curve), respectively. ADHM administrative-driven hierarchical management, AF atrial fibrillation, CI confidence interval, CV cardiovascular, HR hazard ratio, MI myocardial infarction, TIA transient ischemic attack, SE systemic embolism, UC usual care. Time to outcome events were assessed by the Kaplan-Meier method and log-rank test and presented as HR and 95% CI. Two-sided *P* value was adopted. Source data are provided as a Source Data file.

formulated a follow-up plan, provided health education, and generated a referral suggestion if deemed necessary. The detailed implementation strategies of the ACC-to-ABC pathway were shown in the Supplementary Table 6.

The UC cohort was propensity score-matched to the ADHM cohort at a 1:5 ratio from the AF population in the other 13 districts of Shanghai using the SHMHC database. The propensity scores were constructed using the following variables: age, sex, hypertension, diabetes, history of stroke, SE, periphery artery disease, MI, valvular heart disease, chronic obstructive pulmonary disease (COPD), obstructive sleep apnea-hypopnea syndrome (OSHAS), history of major bleeding, cancer, renal deficiency, and liver deficiency. We conducted propensity score matching using a caliper matching protocol (sampling without replacement), and the caliper width was set as 0.02. The characteristics before and after the propensity score matching were shown in the Supplementary Table 7. The index date of the UC cohort corresponded to the date of the ADHM enrollment of the matched patients. Patients in the UC cohort were only observed through database monitoring without any intervention; therefore, they had full autonomy to visit any medical institutions on demand and were cared under the current medical routine, and their identity information was kept confidential from the research team.

### Shanghai AF management platform
The Shanghai AF Management Platform was designed to facilitate the implementation of the ACC-to-ABC pathway in the ADHM cohort. The electronic platform included three ports, i.e., the GP port, the patient port, and the administrative port, facilitating GPs' management, patient education, and administrative supervision, respectively. The detailed information of the platform was shown in Supplementary methods and Supplementary Fig. 4–6.

### Medications and procedures
The medications, procedures, medical costs, and outcome events (except death) were extracted from the SHMHC database, which was government-issued and contained all sources of medical records from 2346 medical institutions in Shanghai since 2015, including inpatient, outpatient, and emergency visits at all CHCs and higher-level hospitals[2]. It also contained medical records of Shanghai residents with government-issued health insurance with allopatry medical activities.

In both groups, the prescriptions for analyses were identified using codes obtained from the Sunshine Medical Procurement All-In-One, a system of centralized purchasing of drugs (including different trade names) covering all hospitals in Shanghai (http://www.smpaa.cn/). Baseline medication use was established by scanning prescription records in the SHMHC database from the 12 months prior to enrollment. Specifically, "regular anticoagulation" was defined as having at least three prescriptions for oral anticoagulants in 12 months. However, the prescription records might not confirm actual medication intake. Procedures, including catheter ablation, LAAC, and cardioversion, were extracted based on the operation codes from the SHMHC database (Supplementary Table 8).

### Referral
Currently, as the electronic systems and referral process among different medical institutions are not fully integrated in China, the referral

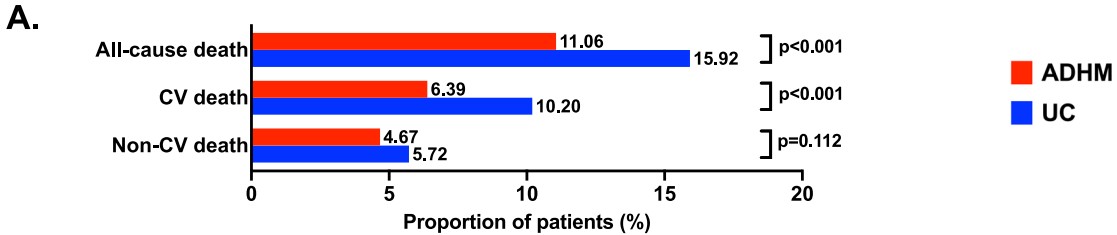

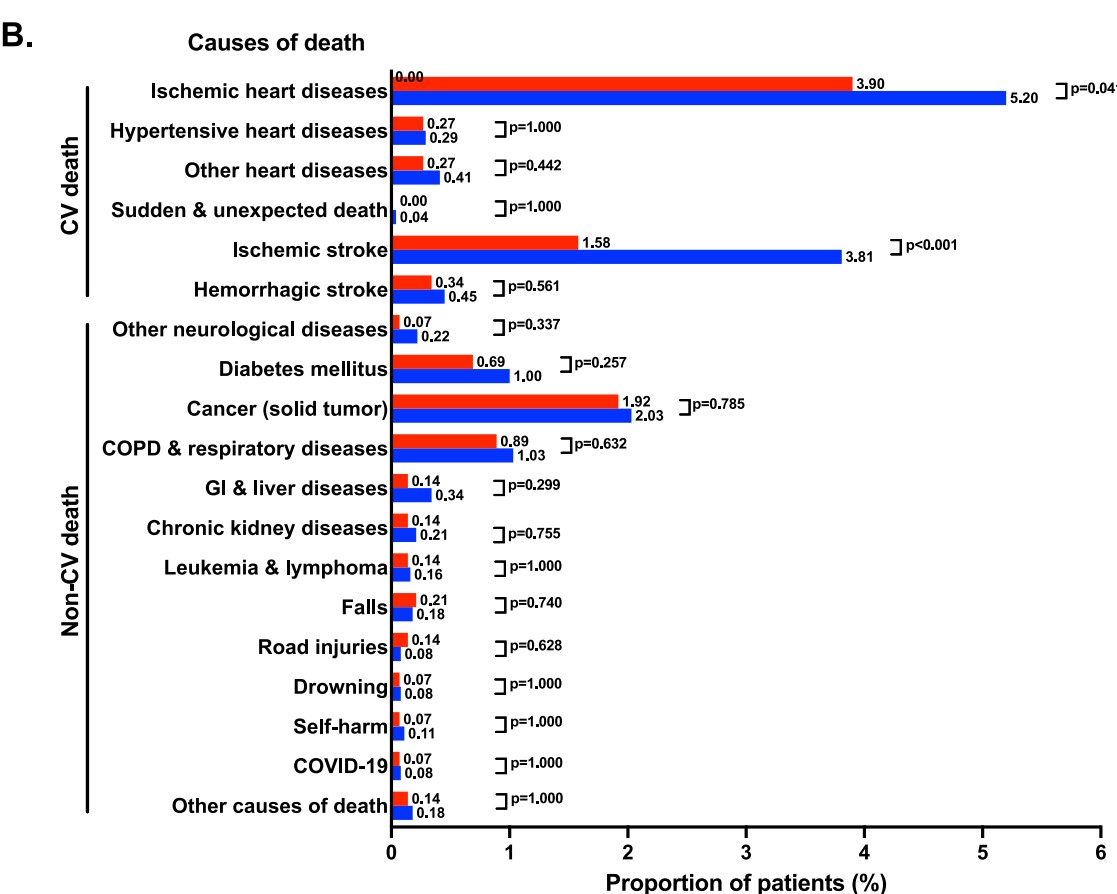

**Fig. 5 | Causes of death.** Shown are the death (**A**) and causes of death (**B**). The numbers above and within the bars indicate the proportion and number of patients who died during the follow-up period, respectively. ADHM administrative-driven hierarchical management; COPD chronic obstructive pulmonary disease; GI gastrointestinal; UC = usual care. The χ2 or Fisher exact tests were applied. Two-sided P value was adopted. Source data are provided as a Source Data file.

process is usually managed without unified records. Therefore, the exact number of referrals could to be determined. As an alternative, taking advantage of the SHMHC database, the times of patients' visits to CHCs versus higher-level hospitals with AF diagnoses were determined to evaluate patients' preferences for healthcare facilities before and after the enrollment. As each prescription could be for up to three months' supply of medication, the "regular" visit to CHCs was defined as ≥4 times visits to CHCs within 12 months.

**Events and follow-up**

The follow-up period was set at 30 months (914 days) post-enrollment. The primary outcome was the composite of CV death, ischemic stroke, SE, MI, major bleeding, and acute HF. The secondary outcome was set as the components of the primary outcomes, all-cause death, and non-CV death. The event definitions were shown in Supplementary Table 9. Outcome events were initially identified through ICD-10 code scanning and then verified by medical records, laboratory tests, and imaging data from two government-issued databases, i.e., the SHMHC database and the Shanghai CDC database. The reason for using this method for

follow-up was that when comparing GPs' follow-up and database scanning at 6 months, despite high consistency, the database screening method showed higher sensitivity due to lower rates of lost to follow-up (Supplementary Fig. 7).

Non-death events (including ischemic stroke, SE, MI, major bleeding, and acute HF) were obtained from the SHMHC database, which contains all sources of medical records of Shanghai residents with government-issued medical insurance. The events of death were obtained from the Shanghai CDC database, as at least 70% of deaths in China occurred outside hospitals without medical records of death[18]. The information on death was aligned and cross-referenced by the Bureaus of Public Security and Civil Affairs, ensuring its timeliness, completeness, and accuracy. The classification of death was shown in Supplementary Table 10.

In addition, the medical costs were obtained from the SHMHC database, which were further categorized into the cost of outpatient, inpatient, or emergency department, in CHCs, and in higher-level hospitals, etc. Cost-effective analysis was performed by calculating the medical cost per patient per survival day.

**Table 2 | Medical costs per patient per survival day**

| Cost per patient per survival day (Chinese Yuan[a]) | Total (n = 8730) | ADHM cohort (n = 1455) | UC cohort (n = 7275) | P value |
|---|---|---|---|---|
| Total costs | 255 ± 593 | 213 ± 266 | 263 ± 638 | <0.001 |
| Total costs in CHCs | 53 ± 204 | 57 ± 177 | 52 ± 209 | 0.360 |
| Total costs in hospitals[†] | 202 ± 531 | 156 ± 191 | 211 ± 575 | <0.001 |
| Total outpatient costs | 47 ± 93 | 60 ± 69 | 45 ± 98 | <0.001 |
| Outpatient costs in CHCs | 22 ± 72 | 35 ± 61 | 20 ± 74 | <0.001 |
| Outpatient costs in hospitals | 25 ± 57 | 25 ± 28 | 25 ± 62 | 0.998 |
| Total inpatient/ED costs | 207 ± 578 | 153 ± 255 | 218 ± 623 | <0.001 |
| Inpatient/ED costs in CHCs[‡] | 31 ± 192 | 22 ± 167 | 33 ± 197 | 0.031 |
| Inpatient/ED costs in hospitals | 176 ± 519 | 131 ± 182 | 185 ± 562 | <0.001 |

[a]Average exchange rate in 2023: 1 Chinese Yuan=0.142 US Dollars.
[†]Hospitals refers to higher-level hospitals.
[‡]A part of CHCs have emergency and inpatient ward.
*ADHM* administrative-driven hierarchical management, *CHC* community healthcare center, *ED* emergency department, *UC* usual care. The independent Student's *t*-tests and two-sided *P* value were adopted. Source data are provided as a Source Data file.

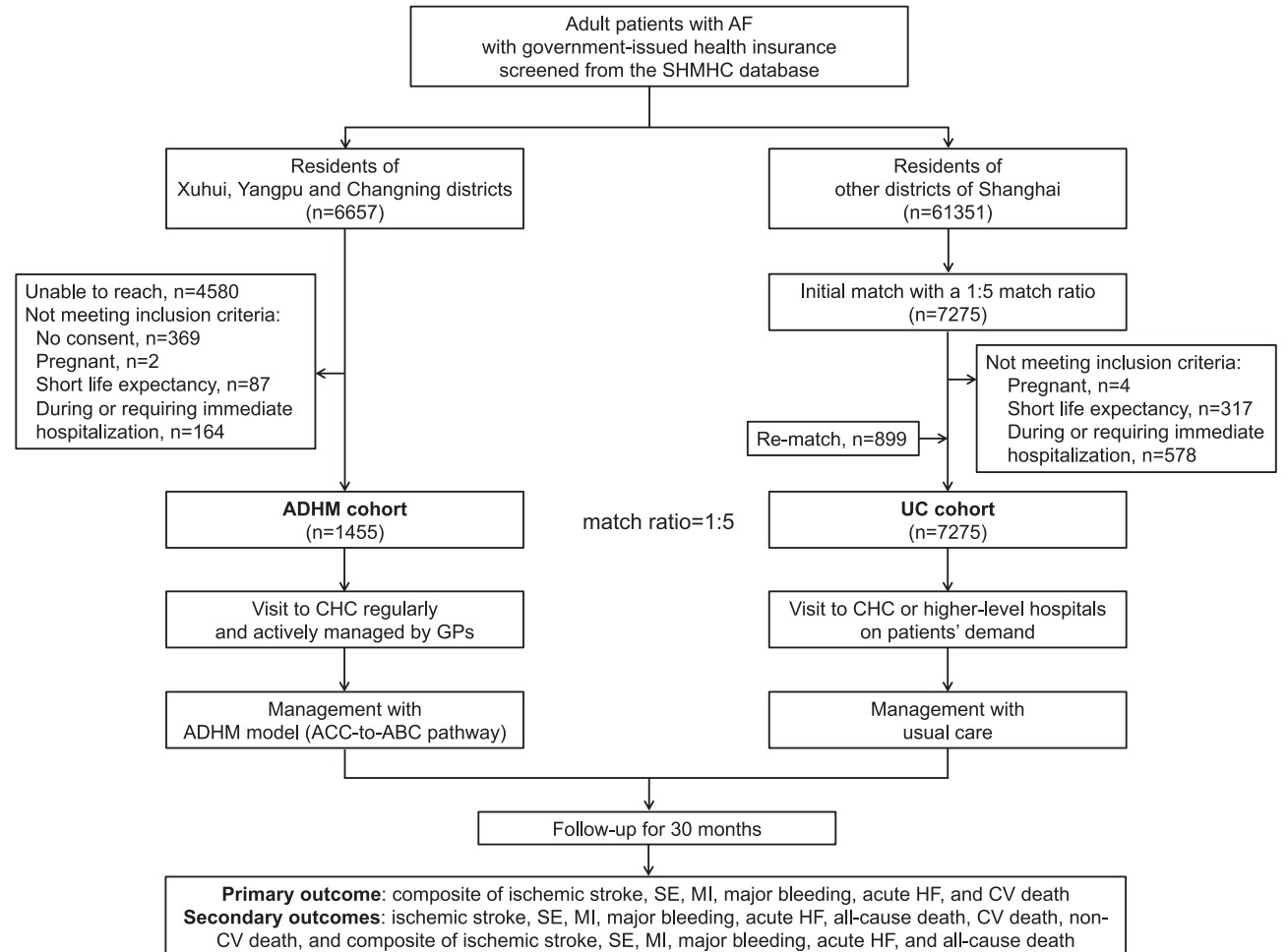

**Fig. 6 | Flowchart of patient enrollment and follow-up.** ADHM administrative-driven hierarchical management, AF atrial fibrillation, GP general practitioners CV cardiovascular, CHC community healthcare center, UC usual care.

## Statistical analyses

Data were shown as n (%) for categorical variables and as mean ± SD (n) for continuous variables. The independent Student's t-test or Mann-Whitney U test was adopted, as appropriate. Categorical data were expressed as counts and percentages and compared between groups by χ2 or Fisher exact tests. Time to outcome events were assessed by the Kaplan-Meier method and log-rank test and presented as hazard ratio (HR) and 95% confidence interval (CI). Univariate and multivariate analyses of predictors of the primary outcome were assessed with the employment of a Cox hazard regression model. The significance level threshold for entry and exit of independent variables into the multivariate model was set at 0.10. Two-sided $P < 0.05$ was considered significant. Data collection and statistical analyses were performed using Python Version 3.12.0 (Python Software Foundation, OR, USA), SPSS

Version 27.0 (IBM Corp., NY, USA), and Prism 9 (GraphPad Software, CA, USA).

## Reporting summary
Further information on research design is available in the Nature Portfolio Reporting Summary linked to this article.

## Data availability
The data supporting the findings described in this manuscript are available in the article and in the Supplementary Information and from the corresponding authors upon request. The data are provided in the Source Data file. Source data are provided with this paper.

## Code availability
The code that supports the findings is available at https://doi.org/10.5281/zenodo.17012225.

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

## Acknowledgements

This study was supported by Shanghai Hospital Development Center (SHDC2020CR2026B, YGL), Shanghai Municipal Health Commission (20224Y0078, MC), Shanghai Municipal Science and Technology Commission (22Y11909900, MY), State Key Program of National Natural Science Foundation of China (82130009, YGL), National Natural Science Foundation of China (82070515, YGL) and Shanghai Leading Talent Plan 2020 (YGL).

## Author contributions

M.C., L.L., H.W., Q.W. and Y.G.L. conceived and designed the study. X.C., C.W., T.X., W.T., L.L., H.W., Q.W. and Y.G.L. supervised the data collection. M.C., M.Z., Y.Y., P.L., Y.F., X.Z., Y.W. and P.C.Y. analyzed and interpreted the data. M.C. drafted the manuscript. All authors revised the manuscript for important intellectual content and approved the final submitted version. M.C., M.Y., J.S., Q.W. and Y.G.L. accessed and verified the data. M.C., H.W., Q.W. and Y.G.L. had full access to all the data in the study and had final responsibility for the decision to submit for publication. All authors vouch for the completeness and accuracy of the data and for the fidelity of the trial to the protocol.

## Competing interests
The authors declare no competing interests.
