## [Transparent Peer Review file · Nature Communications]

Administrative-driven hierarchical management of atrial fibrillation on cardiovascular events: a prospective matched cohort study

Corresponding Author: Dr Yi-Gang Li

Version 0:

Reviewer comments:

Reviewer #1

(Remarks to the Author)

In this manuscript, the authors aimed to investigate the impact of ADHM of AF on cardiovascular events. It was an observational study using three designated districts of Shanghai, China, to construct the ADHM cohort, matched to the UC cohort at a 5:1 ratio. In brief, the authors found that the ADHM model was effective in reducing cardiovascular events and death in individuals with AF.

Comments:

1. Although the authors declared the effectiveness of the ADHM model, the non-randomized study design presented possible biases. How did the authors eliminate the potential co-founders?
2. Following the abovementioned question, despite matching for age and sex, other possible confounding factors such as socioeconomic status, access to healthcare, or comorbidities were not sufficiently analyzed.
3. The study coincided with the COVID-19 pandemic period, which likely influenced healthcare access and outcomes. Could the authors comment on this issue?
4. Could the authors provide long-term outcomes?
5. Were there any differences regarding OAC or AAD use between these two strategies?
6. What measures did the three designated districts take to achieve better treatment quality?
7. Could the authors mention more about the cost-effectiveness in the ADHM

Reviewer #2

(Remarks to the Author)

The present study is interesting, although not very representative for other countries around the globe.

The presented concept is new.

Nevertheless, questions remain:

The authors must provide data about the AF pattern (persistent, permanent or paroxysmal AF?) in the two groups. AF pattern has a substantial impact on outcome (see Goette et al JACC 2022)

The overall rate of anticoagulation is rather low and not different in the two groups. Thus, how did the treatment protocol changes these treatment parameter to explain the observed differences in outcome?

What is the driving factor for the observed difference in outcome? How many patients had sinus rhythm at the end of the study in the two groups?

A multivariate analysis should be presented to determine the most significant effect on outcome for the two groups.

Reviewer #3

(Remarks to the Author)

This paper evaluates the impact of the Administrative-Driven Hierarchical Management (ADHM) model on cardiovascular outcomes in atrial fibrillation (AF) patients within a non-mandatory hierarchical healthcare system in Shanghai, China. The study presents a robust methodology, leveraging government policy interventions, primary care incentives, digital health integration, and bidirectional referral networks to optimize AF management. The findings indicate a significant reduction in cardiovascular events and mortality, alongside lower medical costs. While the study provides valuable insights, there are methodological concerns, potential biases, and limitations in generalizability that should be addressed before publication.

1. The study relies on a matched cohort design rather than a randomized controlled trial (RCT). While age and sex matching ensure some balance, there remains a high potential for residual confounding. Relatedly, the lack of randomization means that selection bias is a concern—patients in the ADHM group may be more engaged or have different baseline characteristics beyond those measured. The authors should perform propensity score matching (PSM) or weighting (PSW) to strengthen causal inference.

2. There could be other sensitivity analyses added to further alleviate the unmeasured confounder issue. For instance, the study does not address unmeasured confounders, such as socioeconomic status, severity of AF at baseline, and other health behaviors. The authors should consider all of these measures (if available). The authors should at least adjust for some of these covariates or perform E-value analysis (on top of the propensity score analysis) to estimate the magnitude of an unmeasured confounder required to nullify the observed associations.

3. The study period coincides with the COVID-19 pandemic, particularly during China's shift from "Zero-COVID" to full reopening (late 2022 - early 2023). The sharp drop in survival curves during this period suggests excess COVID-19-related mortality, which could bias overall mortality results. The authors can consider conducting sensitivity analysis excluding COVID-19 deaths and/or stratify results by pre- and post-pandemic periods.

4. The authors claim reduced visits to higher-level hospitals, but do not quantify referral rates explicitly. Referral data are inferred indirectly from hospital visits, but lack precise tracking of referrals between CHCs and hospitals. If possible, the authors should provide direct referral rate data rather than inferring from hospital visits. Also, did CHC-based management result in delayed escalation for critical cases?

5. As a major statistical problem, the primary endpoint (composite cardiovascular events) uses a Cox proportional hazards model, but does not account for competing risks (e.g., non-cardiovascular deaths). Given the high mortality rate, competing risk regression (Fine-Gray model) should be considered, if there's no way to justify the use of all cause mortality as the primary outcome.

6. The increase in anticoagulation use (88.7% vs. 59.5%) is cited as a key mechanism for improved outcomes. However, adherence data rely on prescription records, which do not confirm actual medication intake. The authors can consider using proportion of days covered (PDC) as a proxy for adherence. Also, the potential prescription fill bias (i.e., patients filling prescriptions but not taking medications) needs to be elaborated.

7. In the context of China, Shanghai is quite an outlier in terms of economic development. The Shanghai-based intervention may not generalize to rural China or other LMICs without government-driven healthcare.

8. The role of financial incentives in physician behavior is not fully explored—Did incentives drive overprescription or unnecessary procedures?

9. The follow up term can be longer? 21 months is kind of in btw short and long terms, making it unclear whether benefits will persist.

10. Another major problem is that the authors report lower costs in the ADHM cohort, but what is the cost-effectiveness ratio? The authors did not provide a breakdown of cost drivers nor consider indirect costs (e.g., patient travel, productivity loss). A cost expert needs to be involved to assess this with the more standard cost-assessment frameworks.

11. The KM curves in Figure 5 lack clear separation. Please consider adding time-dependent hazard ratios.

12. For Table 1, please highlight standardized mean differences (SMDs) to show balance across cohorts.

13. Some of the long descriptive texts here and there (e.g., Shanghai AF Management Platform) should be shortened and moved to appendix.

Version 1:

Reviewer comments:

Reviewer #1

(Remarks to the Author)

The authors have responded to all the comments satisfactorily.

Reviewer #2

(Remarks to the Author)

The response letter with specific answers was not uploaded

Reviewer #3

(Remarks to the Author)

Most of my comments are addressed in this revision. It is understandable that data limitation make some of my comments unaddressable.

REVIEWER COMMENTS

Reviewer #1 (Remarks to the Author):

In this manuscript, the authors aimed to investigate the impact of ADHM of AF on cardiovascular events. It was an observational study using three designated districts of Shanghai, China, to construct the ADHM cohort, matched to the UC cohort at a 5:1 ratio. In brief, the authors found that the ADHM model was effective in reducing cardiovascular events and death in individuals with AF.

Comments:

1. Although the authors declared the effectiveness of the ADHM model, the non-randomized study design presented possible biases. How did the authors eliminate the potential co-founders?

Response: Thanks for the insightful comment. The implementation of the ADHM model requires the administrative orders and official documents issued by the local government. When designing the study, the Shanghai Municipal Health Commission designated three administrative districts to imply the model. Therefore, the study could not be performed in form of randomized study, so we fully agree there would be possible bias. In the revised manuscript, to minimize the bias, we performed propensity score matching to construct the new UC cohort instead of the matching of only sex and age. The propensity scores were constructed using age, sex and 13 comorbidities, including hypertension, diabetes, stroke, systemic embolism, periphery artery disease, myocardial infarction, valvular heart disease, chronic obstructive pulmonary disease, obstructive sleep apnea-hypopnea syndrome, major bleeding, cancer, renal deficiency and liver deficiency (Supplementary Table 7), to recude the potential impact of confounders. The results in the revised manuscript were consistent with previously reported data that ADHM model reduced cardiovascular events in patients with AF.

Supplementary Table 7. Characteristics before and after propensity score matching.

	Before propensity score matching	After propensity score matching
--	----------------------------------	---------------------------------

Characteristics	ADHM cohort	control	SMD	P value	ADHM cohort	UC cohort	SMD	P value
Female sex (%)	45.9	51.7	0.097	<0.001	45.9	45.9	0.000	1.000
Age (years)	76.0	79.0	-0.247	<0.001	76.0	76.0	0.006	0.848
Hypertension (%)	92.0	95.7	0.097	<0.001	92.0	93.0	0.016	0.201
Diabetes (%)	54.6	50.6	0.079	0.002	54.6	54.5	0.003	0.908
Stroke/TIA (%)	51.4	55.4	-0.081	0.002	51.4	51.2	0.031	0.590
Systemic embolism (%)	3.9	4.6	-0.033	0.167	3.9	3.3	0.032	0.270
Peripheral artery disease (%)	4.3	3.5	0.044	0.096	4.3	3.4	0.048	0.091
Myocardial infarction (%)	3.4	5.9	-0.106	<0.001	3.4	3.0	0.021	0.471
Valvular heart disease (%)	3.5	3.0	0.027	0.305	3.5	3.3	0.014	0.630
COPD (%)	22.5	28.2	-0.126	<0.001	22.5	22.2	0.007	0.800
OSHAS (%)	1.1	0.6	0.061	0.021	1.1	0.7	0.042	0.144
Major bleeding (%)	11.9	16.4	-0.121	<0.001	11.9	11.0	0.029	0.308
History of cancer (%)	12.3	18.4	-0.159	<0.001	12.3	12.4	-0.004	0.896
Renal deficiency (%)	5.7	5.1	0.026	0.331	5.7	5.0	0.034	0.239
Liver deficiency (%)	15.9	15.8	0.002	0.929	15.9	16.1	-0.007	0.815

2. Following the abovementioned question, despite matching for age and sex, other possible confounding factors such as socioeconomic status, access to healthcare, or comorbidities were not sufficiently analyzed.

Response: In the revised manuscript, we constructed a new UC cohort using propensity score matching to further reduce the potential confounding factors, including age, sex and 13 related comorbidities (Supplementary Table 7). Currently, the socioeconomic status was not available in the database, and we added this concern in the limitation (page 11, line 3-4). The access to healthcare (frequency of visit to community healthcare center or higher-level hospitals) was similar before the enrollment between two groups (Figure 2).

Figure 2. Patients visiting CHC and higher-level hospitals.

3. The study coincided with the COVID-19 pandemic period, which likely

influenced healthcare access and outcomes. Could the authors comment on this issue?

Response: Indeed, the impact of the COVID-19 on this study is undeniable. The follow-up period coincided with the shift in China's COVID-19 policies from the "dynamic zero-COVID" to full reopening, resulting in a large wave of Omicron variant infections and excess mortalities within a focal period (approximately from November 2022 to January 2023). In the revised manuscript, we performed analyses before/during and after the shift in China's COVID-19 policies (the cut-off date was set as January 31, 2023). As shown in the revised Supplementary Figure 2, the results were consistent before/during and after the Omicron outbreak. In addition, since participants in both groups lived in the same city, we reasonably assumed that such impact was similar on both groups. We have addressed the concern in the limitations. It reads: "First, the follow-up period coincided with the shift in China's COVID-19 policies from the "dynamic zero-COVID" to full reopening, resulting in a large wave of Omicron variant infections and excess mortalities within a focal period (approximately from November 2022 to January 2023). Consequently, a sharp drop on the survival curve was observed during the corresponding period. During the outbreak, patients' behaviors in healthcare seeking might termed irrational and GPs themselves were also widely infected, leading to an interruption in the ADHM implementation. Additionally, anti-SARS-CoV-2 medications (such as nirmatrelvir/ritonavir) cannot be used in conjunction with rivaroxaban or amiodarone, and dosage adjustment for other anticoagulants was also required, leading to interruptions or deviations from standard therapies. Hence, the impact of the COVID-19 on this study is undeniable. However, as the Kaplan-Meier estimates of the primary outcome showed consistency before/during and after the Omicron outbreak, such impact was reasonably assumed similar between both groups." (page 10, line 8-20).

Supplementary Fig. 2. Outcomes of patients with AF before/during and after the shift in China's COVID-19 policies.

Shown are Kaplan-Meier estimates of the composite of CV death, ischemic stroke, SE, MI, major bleeding, and acute HF (primary endpoint), and the composite of all-cause death, ischemic stroke, SE, MI, major bleeding, and acute HF, before (A, B) and after (C, D) January 31, 2023.

4. Could the authors provide long-term outcomes?

Response: As you suggested, we presented 30 months follow-up results in the revised manuscript (21 months in the previous version). The new results were mostly consistent as the data derived from 21 months follow-up.

5. Were there any differences regarding OAC or AAD use between these two strategies?

Response: Yes, the results are analyzed and presented in revised Figure 3 now. During the follow-up, the ADHM cohort had higher rates of OAC and AAD prescription than the

UC cohort.

Figure 3. Proportion of patients receiving stroke prevention and symptom control therapies.

Proportion of patients receiving at least one anticoagulant prescription (A), regular anticoagulation (B), LAAC (C), antiarrhythmic drugs (D), and catheter ablation (E) at baseline and during follow-up. ADHM= administrative-driven hierarchical management; LAAC= left atrial appendage closure; UC= usual care.

6. What measures did the three designated districts take to achieve better treatment quality?

Response: The measures to implement ADHM model in the three designated districts were shown in the revised Supplementary Table 6 now.

Supplementary Table 6. Implementation strategies of the ADHM model.

Shanghai Municipal Health Commission  • Launch of the White Paper on Standardized Management of Atrial Fibrillation in Shanghai • Issuing administrative orders to the Health Commission of the pilot districts (Xuhui, Yangpu, and Changning districts) implementing the ADHM model. • Semi-annual conferences on work • Establishment of performance incentives for CHCs and GPs • Access to the administrative port of the Shanghai AF Management Platform (hereinafter referred to as “the Platform”) for data monitoring and supervision
District Health Commission

- Issuing administrative orders to subordinate CHCs implementing the ADHM model.
- Implementation of performance incentives for CHCs
- Quarterly conference on work
- Ensuring the availability and stockpile of AF-related drugs in CHCs
- Access to the administrative port of the Platform for data monitoring and supervision
- Strengthening the regional healthcare network to facilitate referrals between subordinate CHCs and nearby higher-level hospitals

Medical Association*

- Guideline recommendation related to AF management
- Public health policy consultation
- Access to the administrative port of the Platform for data monitoring and supervision
- Knowledgebase updating for the Platform

Shanghai AF Center Union#

- Member hospitals of Shanghai AF Center Union accepting referrals from CHCs: interventional therapy, complications, and critical conditions
- Training for GPs on AF management
- Training for GPs on the use of the Platform
- Access to the administrative port of the Platform for data monitoring and supervision

CHCs

- Designation of at least one GP in charge of ADHM model implementation
- Implementation of performance and non-performance-based incentives for GPs. Note that the financial incentives are not tied to drug prescriptions or referrals to cardiac procedures but rather to data entry on the Shanghai AF management platform and regular patient follow-ups.
- Ensuring the availability and stockpile of AF-related drugs and examinations such as international normalized ratio
- Authorizing the installation of the Platform in the hospital network environment
- Access to the GP port of the Platform
- Organization of training conferences
- Organization of patient education activities

GPs

- Receiving training on AF management at the start of the intervention
- Receiving training on the use of the Platform at the start of the intervention
- Conferences every 6 months: sharing experience, discussing cases, and additional training on topics based on existing questions
- Access to the GP port of the Platform
- Enrollment of patients with AF
- Risk assessment: CHA₂DS₂-VASc, HAS-BLED, and EHRA scores
- Management: anticoagulation, symptom control, and management of cardiovascular risk factors, comorbidities, and complications, including hypertension, diabetes, coronary artery disease, chronic obstructive pulmonary disease, chronic kidney disease, etc.
- Follow-up: Follow-up plan, in-person follow-up, telephone follow-up
- Two-way referral: upward referral and receiving downward referral from upper-level hospitals
- Patient education: in-person education, push messages to the patient port through the Platform
- With the assistance of intelligent-generated, patient-individualized recommendations for treatment, follow-up plan and referral from the Platform
- Consultation with the specialist physician from the Shanghai AF Center Union for therapeutic decision-making if needed

Patients with AF  • Signing of the informed consent • Access to the patient port of the Platform • Receiving treatment including anticoagulation, symptom control, and management of cardiovascular risk factors, comorbidities, and complications • Receiving push messages from the GPs regarding personalized AF-related risk scores, drug instruction, follow-up plan, and health education
Family members of patients with AF  • Access to the patient port of the Platform if authorized • Receiving push messages from the GPs regarding personalized AF-related risk scores, drug instruction, follow-up plan, and health education

7. Could the authors mention more about the cost-effectiveness in the ADHM.

Response: We calculate the medical cost per patient per survival day in the revised manuscript (revised Table 2). The ADHM model saved nearly 50 Chinese yuan per patient per survival day compared with the UC cohort. Thanks for your insightful comments again.

Table 2. Medical costs per patient per survival day.

Cost per patient per survival day (Chinese Yuan*)	Total (n=8730)	ADHM cohort (n=1455)	UC cohort (n=7275)	P value
Total costs	255 ± 593	213 ± 266	263 ± 638	<0.001
Total costs in CHCs	53 ± 204	57 ± 177	52 ± 209	0.360
Total costs in hospitals [†]	202 ± 531	156 ± 191	211 ± 575	<0.001
Total outpatient costs	47 ± 93	60 ± 69	45 ± 98	<0.001
Outpatient costs in CHCs	22 ± 72	35 ± 61	20 ± 74	<0.001
Outpatient costs in hospitals	25 ± 57	25 ± 28	25 ± 62	0.998
Total inpatient/ED costs	207 ± 578	153 ± 255	218 ± 623	<0.001
Inpatient/ED costs in CHCs [‡]	31 ± 192	22 ± 167	33 ± 197	0.031
Inpatient/ED costs in hospitals	176 ± 519	131 ± 182	185 ± 562	<0.001

* Average exchange rate in 2023: 1 Chinese Yuan=0.142 US Dollars.

[†] Hospitals refers to higher-level hospitals.

[‡] A part of CHCs have emergency and inpatient ward.

Reviewer #2 (Remarks to the Author):

The present study is interesting, although not very representative for other countries around the globe. The presented concept is new. Nevertheless, questions remain:

The authors must provide data about the AF pattern (persistent, permanent or paroxysmal AF?) in the two groups. AF pattern has a substantial impact on outcome (see Goette et al JACC 2022).

Response: Thanks for the insightful comments. We agree the AF pattern is very important factors regarding cardiovascular outcomes. However, the current database adopted ICD-10 codes which did not differentiate persistent, permanent or paroxysmal AF (ICD-11 codes did, but not available in the current database). We have added the concern in the limitations (page 11, line 1-3). It reads: "Fourth, the pattern of AF, i.e., paroxysmal, persistent, or permanent AF, was not able to be differentiate due to the adoption of ICD-10 codes in the database rather than ICD-11 codes."

The overall rate of anticoagulation is rather low and not different in the two groups. Thus, how did the treatment protocol changes these treatment parameter to explain the observed differences in outcome?

Response: Thanks for pointing out this important issue. Yes, overall rate of anticoagulation was relatively low and similar between the two groups at baseline (approximately 40%). The rate was significantly increased after ADHM intervention (78.4%), compared with the UC cohort (46.3%), as shown in revised Figure 3A, 3B. Multivariate alaysis showed that regular anticoagulation was the independent protective factor of cardiovascular outcomes in the ADHM cohort (Supplementary Table 2). This point is clarified in the revised manuscript now.

Figure 3A,B Propotion of patients receicing anticoagulation.

Proportion of patients receiving at least one anticoagulant prescription (A), regular anticoagulation (B) at baseline and during follow-up.

What is the driving factor for the observed difference in outcome? How many patients had sinus rhythm at the end of the study in the two groups?

Response: Thanks for pointing out these important issues. These data are not available based on the data scan from ICD coding, and we described this study limitation in the revised manuscript now (page 11, line 3-4). It reads: “The status of rhythm (AF and its burden or sinus rhythm) at the end of the study, socioeconomic status and other health behaviors was also not available in the database.” Multivariate analyses were performed in the entire cohort, the ADHM cohort and the UC cohort, and the results were presented in the revised Supplementary Table 1, 2 and 3, respectively, showing the different management model (ADHM vs UC) was the independent driving factors of cardiovascular outcomes in patients with AF.

A multivariate analysis should be presented to determine the most significant effect on outcome for the two groups.

Response: Thanks and we followed the helpful comments. The results presented in the Supplementary Table 1, 2 and 3, and results section (page 7, line 1-8) now.

Supplementary Table 1. Predictors of the primary outcome in patients in both ADHM and UC groups.

Variables	Univariate		Multivariate	
	HR (95% CI)	P value	HR (95% CI)	P value
Baseline characteristics				
Age (years)	1.059 (1.052-1.065)	<0.001	1.053 (1.041-1.065)	<0.001

Female sex	1.286 (1.161-1.424)	<0.001	1.203 (0.972-1.490)	0.090
Hypertension	2.457 (1.894-3.188)	<0.001	1.176 (0.788-1.754)	0.427
Diabetes	1.381 (1.245-1.532)	<0.001	1.212 (0.974-1.509)	0.085
History of stroke/TIA	2.325 (2.088-2.588)	<0.001	1.779 (1.212-2.611)	0.003
History of SE	1.548 (1.128-1.885)	0.004	1.099 (0.839-1.440)	0.493
History of heart failure	1.793 (1.614-1.992)	<0.001	1.452 (1.163-1.812)	0.001
Periphery artery disease	1.690 (1.320-2.163)	<0.001	1.379 (1.065-1.787)	0.015
History of MI	1.458 (1.112-1.911)	0.006	1.392 (0.991-1.955)	0.057
Valvular heart disease	0.852 (0.647-1.120)	0.251		
COPD	1.289 (1.145-1.451)	<0.001	0.923 (0.813-1.048)	0.217
History of major bleeding	1.399 (1.202-1.629)	<0.001	0.940 (0.656-1.347)	0.738
History of cancer	0.953 (0.815-1.115)	0.549		
History of renal deficiency	1.971 (1.609-2.415)	<0.001	1.327 (0.899-1.959)	0.154
History of liver deficiency	1.556 (1.367-1.771)	<0.001	1.199 (0.843-1.705)	0.313
CHA2DS2-VASc score	1.355 (1.315-1.396)	<0.001	0.907 (0.752-1.093)	0.306
HAS-BLED score	1.658 (1.569-1.752)	<0.001	1.164 (0.842-1.609)	0.359
During follow-up				
UC compared to ADHM	1.674 (1.435-1.952)	<0.001	1.838 (1.556-2.173)	<0.001
Regular anticoagulant prescription	0.872 (0.788-0.966)	0.009	0.962 (0.862-1.073)	0.487
Antiarrhythmic drug prescription	0.998 (0.899-1.108)	0.966		
Rate control drug prescription	1.028 (0.921-1.148)	0.616		
Statin prescription	0.975 (0.838-1.135)	0.747		
RAASi prescription	1.024 (0.912-1.151)	0.685		
MRA prescription	1.075 (0.917-1.262)	0.372		
SGLT2i prescription	1.058 (0.832-1.346)	0.643		
Catheter ablation	1.173 (0.928-1.483)	0.181		
LAAC	1.347 (0.942-1.926)	0.102		
Visit CHCs≥4 times per year	0.912 (0.824-1.011)	0.078	1.246 (1.113-1.396)	<0.001
Times of visit to higher level hospitals	1.005 (0.984-1.026)	0.645		

Supplementary Table 2. Predictors of the primary outcome in patients in the ADHM group.

Variables	Univariate		Multivariate	
	HR (95% CI)	P value	HR (95% CI)	P value
Baseline characteristics				
Age (years)	1.099 (1.079-1.120)	<0.001	1.062 (1.028-1.097)	<0.001
Female sex	1.426 (1.069-1.902)	0.016	0.733 (0.379-1.419)	0.357
Hypertension	4.223 (1.703-10.469)	0.002	2.922 (0.777-10.985)	0.112
Diabetes	1.572 (1.167-2.117)	0.003	0.695 (0.351-1.378)	0.298
History of stroke/TIA	2.561 (1.877-3.494)	<0.001	1.454 (0.465-4.546)	0.520
History of SE	1.089 (0.508-2.332)	0.827		
History of heart failure	2.903 (2.111-3.992)	<0.001	1.010 (0.511-1.996)	0.978
Periphery artery disease	1.089 (0.545-2.178)	0.808		
History of MI	2.104 (1.097-3.034)	0.025	0.904 (0.351-2.329)	0.834

Valvular heart disease	1.395 (0.688-2.827)	0.356		
COPD	1.790 (1.305-2.453)	<0.001	1.102 (0.778-1.561)	0.585
History of major bleeding	1.712 (1.156-2.536)	0.007	3.199 (1.005-10.184)	0.049
History of cancer	1.325 (0.880-1.993)	0.177		
History of renal deficiency	1.740 (1.020-2.966)	0.042	3.353 (0.980-11.469)	0.054
History of liver deficiency	1.615 (1.131-2.307)	0.008	3.624 (1.122-11.701)	0.031
CHA2DS2-VASc score	1.533 (1.400-1.677)	<0.001	1.589 (0.887-2.847)	0.119
HAS-BLED score	1.752 (1.504-2.041)	<0.001	0.401 (0.134-1.200)	0.102
During follow-up				
Regular anticoagulant prescription	0.592 (0.429-0.816)	0.001	0.555 (0.384-0.801)	0.002
Antiarrhythmic drug prescription	1.012 (0.759-1.350)	0.935		
Rate control drug prescription	1.439 (1.070-1.936)	0.016	1.217 (0.874-1.695)	0.244
Statin prescription	1.515 (1.116-2.057)	0.747		
RAASi prescription	1.628 (1.219-1.173)	0.001	1.354 (0.984-1.864)	0.062
MRA prescription	2.465 (1.796-3.383)	<0.001	1.289 (0.899-1.848)	0.167
SGLT2i prescription	1.457 (0.937-2.265)	0.095	1.407 (0.848-2.334)	0.186
Catheter ablation	0.491 (0.288-0.837)	0.009	0.830 (0.468-1.470)	0.522
LAAC	1.403 (0.593-3.323)	0.441		
Visit CHCs \geq 4 times per year	0.576 (0.420-0.791)	0.001	0.971 (0.709-1.329)	0.855
Times of visit to higher level hospitals	0.951 (0.869-1.040)	0.269		

Supplementary Table 3. Predictors of the primary outcome in patients in the UC group.

Variables	Univariate		Multivariate	
	HR (95% CI)	P value	HR (95% CI)	P value
Baseline characteristics				
Age (years)	1.054 (1.047-1.160)	<0.001	1.046 (1.034-1.059)	<0.001
Female sex	1.269 (1.137-1.415)	<0.001	1.248 (0.994-1.566)	0.056
Hypertension	2.291 (1.744-3.011)	<0.001	1.016 (0.666-1.550)	0.940
Diabetes	1.359 (1.216-1.519)	<0.001	1.232 (0.976-1.555)	0.079
History of stroke/TIA	2.306 (2.056-2.587)	<0.001	1.634 (1.087-2.457)	0.018
History of SE	1.591 (1.208-2.097)	0.001		
History of heart failure	1.683 (1.505-1.883)	<0.001	1.010 (0.511-1.996)	0.978
Periphery artery disease	1.869 (1.429-2.444)	<0.001	1.454 (1.101-1.921)	0.008
History of MI	1.376 (1.022-1.853)	0.036	1.323 (0.916-1.911)	0.135
Valvular heart disease	1.148 (0.852-1.547)	0.363		
COPD	1.228 (1.080-1.396)	0.002	0.892 (0.778-1.022)	0.100
History of major bleeding	1.365 (1.157-1.610)	<0.001	0.783 (0.536-1.145)	0.208
History of cancer	1.106 (0.934-1.310)	0.244		
History of renal deficiency	2.051 (1.644-2.559)	<0.001	1.164 (0.770-1.759)	0.473
History of liver deficiency	1.550 (1.348-1.781)	<0.001	1.044 (0.721-1.512)	0.821
CHA2DS2-VASc score	1.335 (1.294-1.378)	<0.001	0.885 (0.725-1.079)	0.228
HAS-BLED score	1.652 (1.557-1.753)	<0.001	1.347 (0.959-1.893)	0.086
During follow-up				

Regular anticoagulant prescription	0.994 (0.891-1.110)	0.919		
Antiarrhythmic drug prescription	1.023 (0.914-1.145)	0.694		
Rate control drug prescription	1.085 (0.959-1.227)	0.195		
Statin prescription	0.964 (0.804-1.156)	0.692		
RAASi prescription	11.023 (0.897-1.167)	0.734		
MRA prescription	0.907 (0.749-1.099)	0.318		
SGLT2i prescription	1.064 (0.795-1.423)	0.678		
Catheter ablation	1.190 (0.288-0.837)	0.009	1.082 (0.801-1.462)	0.607
LAAC	1.830 (1.216-2.754)	0.004	1.914 (1.211-3.023)	0.005
Visit CHCs \geq 4 times per year	1.037 (0.930-1.157)	0.511		
Times of visit to higher level hospitals	0.989 (0.967-1.011)	0.320		

Reviewer #3 (Remarks to the Author):

This paper evaluates the impact of the Administrative-Driven Hierarchical Management (ADHM) model on cardiovascular outcomes in atrial fibrillation (AF) patients within a non-mandatory hierarchical healthcare system in Shanghai, China. The study presents a robust methodology, leveraging government policy interventions, primary care incentives, digital health integration, and bidirectional referral networks to optimize AF management. The findings indicate a significant reduction in cardiovascular events and mortality, alongside lower medical costs. While the study provides valuable insights, there are methodological concerns, potential biases, and limitations in generalizability that should be addressed before publication.

1. The study relies on a matched cohort design rather than a randomized controlled trial (RCT). While age and sex matching ensure some balance, there remains a high potential for residual confounding. Relatedly, the lack of randomization means that selection bias is a concern—patients in the ADHM group may be more engaged or have different baseline characteristics beyond those measured. The authors should perform propensity score matching (PSM) or weighting (PSW) to strengthen causal inference.

Response: We followed the valuable suggestions. In the revised manuscript, propensity score matching was performed which matched with age, sex and 13 comorbidities, including hypertension, diabetes, stroke, systemic embolism, periphery artery disease, myocardial infarction, valvular heart disease, chronic obstructive pulmonary disease, obstructive sleep apnea-hypopnea syndrome, major bleeding, cancer, renal deficiency and liver deficiency (Supplementary Table 7) to construct a new UC cohort. The results are consistent with previously reported data that ADHM model reduced cardiovascular events in patients with AF.

Characteristics before and after propensity score matching.

Characteristics	Before propensity score matching				After propensity score matching			
	ADHM cohort	control	SMD	P value	ADHM cohort	UC cohort	SMD	P value
Female sex (%)	45.9	51.7	0.097	<0.001	45.9	45.9	0.000	1.000

Age (years)	76.0	79.0	-0.247	<0.001	76.0	76.0	0.006	0.848
Hypertension (%)	92.0	95.7	0.097	<0.001	92.0	93.0	0.016	0.201
Diabetes (%)	54.6	50.6	0.079	0.002	54.6	54.5	0.003	0.908
Stroke/TIA (%)	51.4	55.4	-0.081	0.002	51.4	51.2	0.031	0.590
Systemic embolism (%)	3.9	4.6	-0.033	0.167	3.9	3.3	0.032	0.270
Peripheral artery disease (%)	4.3	3.5	0.044	0.096	4.3	3.4	0.048	0.091
Myocardial infarction (%)	3.4	5.9	-0.106	<0.001	3.4	3.0	0.021	0.471
Valvular heart disease (%)	3.5	3.0	0.027	0.305	3.5	3.3	0.014	0.630
COPD (%)	22.5	28.2	-0.126	<0.001	22.5	22.2	0.007	0.800
OSHAS (%)	1.1	0.6	0.061	0.021	1.1	0.7	0.042	0.144
Major bleeding (%)	11.9	16.4	-0.121	<0.001	11.9	11.0	0.029	0.308
History of cancer (%)	12.3	18.4	-0.159	<0.001	12.3	12.4	-0.004	0.896
Renal deficiency (%)	5.7	5.1	0.026	0.331	5.7	5.0	0.034	0.239
Liver deficiency (%)	15.9	15.8	0.002	0.929	15.9	16.1	-0.007	0.815

2. There could be other sensitivity analyses added to further alleviate the unmeasured confounder issue. For instance, the study does not address unmeasured confounders, such as socioeconomic status, severity of AF at baseline, and other health behaviors. The authors should consider all of these measures (if available). The authors should at least adjust for some of these covariates or perform E-value analysis (on top of the propensity score analysis) to estimate the magnitude of an unmeasured confounder required to nullify the observed associations.

Response: In the revised manuscript, we performed propensity score matching which adjusted age, sex and other 13 AF-related comorbidities to reduce the potential confoundings (Supplementary Table 7). And the results were revised accordingly throughout the entire manuscript. The main outcomes remained consistent between the new and previously reported data. However, socioeconomic status, severity of AF at baseline, and other health behaviors were not available in the database, and we stated this study limitation in the revised manuscript now (page 11, line 1-4). It reads: “Fourth, the pattern of AF, i.e., paroxysmal, persistent, or permanent AF, was not able to be differentiate due to the adoption of ICD-10 codes in the database rather than ICD-11 codes. The status of rhythm (AF and its burden or sinus rhythm) at the end of the study, socioeconomic status and other health behaviors was also not available in the database.”

3. The study period coincides with the COVID-19 pandemic, particularly during China's shift from "Zero-COVID" to full reopening (late 2022 - early 2023). The sharp drop in survival curves during this period suggests excess COVID-19-related mortality, which could bias overall mortality results. The authors can consider conducting sensitivity analysis excluding COVID-19 deaths and/or stratify results by pre- and post-pandemic periods.

Response: We followed the valuable suggestion and performed stratified analyses before/during and after the Omicron outbreak during the shift in China's COVID-19 policies (the cut-off date was set as January 31, 2023). As shown in the Supplementary Figure 2, the results consistently favored the ADHM model over the usual care regarding the cardiovascular events in patients with AF.

Outcomes of patients with AF before/during and after the shift in China's COVID-19 policies.

Shown are Kaplan-Meier estimates of the composite of CV death, ischemic stroke, SE, MI, major bleeding, and acute HF (primary endpoint), and the composite of all-cause death, ischemic stroke, SE, MI, major bleeding, and acute HF, before (A, B) and after (C, D) January 31, 2023.

4. The authors claim reduced visits to higher-level hospitals, but do not quantify referral rates explicitly. Referral data are inferred indirectly from hospital visits, but lack precise tracking of referrals between CHCs and hospitals. If possible, the authors should provide direct referral rate data rather than inferring from hospital visits. Also, did CHC-based management result in delayed escalation for critical cases?

Response: Thanks for the insightful comment. We only reported the reduced visits to higher-level hospitals, but did not quantify referral rates explicitly. The reasons are as follows: 1. We have no data on the direct referrals to higher hospitals in the database, since the the electronic systems and referral process among different medical institutions were not integrated in China, and the referral process were usually managed without unified records. 2. In China, patients could directly visit higher hospitals without GP's recommendation on their own willing, even in the ADHM cohort, therefore the CHC-based management unlikely resulted in delayed escalation. We specified this condition in the method section (page 13, line 8-11). It reads: "Currently, as the electronic systems and referral process among different medical institutions were not fully integrated in China, the referral process were usually managed without unified records. Therefore, the exact number of referrals was unable to be determined." Recently, the local governments are working to establish a unified two-way referral network among the entire city.

5. As a major statistical problem, the primary endpoint (composite cardiovascular events) uses a Cox proportional hazards model, but does not account for competing risks (e.g., non-cardiovascular deaths). Given the high mortality rate, competing risk regression (Fine-Gray model) should be considered, if there's no way to justify the use of all cause mortality as the primary outcome.

Response: Thanks for the insightful comment. The composite of cardiovascular events and all-cause death, as well as all-cause death, were compared between groups, and presented in the revised Figure 4B, C, and Supplementary Figure 2.

6. The increase in anticoagulation use (88.7% vs. 59.5%) is cited as a key mechanism for improved outcomes. However, adherence data rely on prescription records, which do not confirm actual medication intake. The authors can consider using proportion of days covered (PDC) as a proxy for adherence. Also, the potential prescription fill bias (i.e., patients filling prescriptions but not taking medications) needs to be elaborated.

Response: Thanks for the helpful comments. We fully agree, however, based on the available dataset, we are unable to obtain related PDC and prescription fill bias data. We use “anticoagulant prescription” instead of “anticoagulant use” in this study. In the revised manuscript, we specified this issue in method section (page 13, line 2-4). It reads: “Specifically, “regular anticoagulation” was defined as having at least three prescriptions for oral anticoagulants in 12 months. However, the prescription records might not confirm actual medication intake.”

7. In the context of China, Shanghai is quite an outlier in terms of economic development. The Shanghai-based intervention may not generalize to rural China or other LMICs without government-driven healthcare.

Response: Thanks and we agree. This point is mentioned in the discussion section now. It reads: “However, as Shanghai is a relatively economically developed city, whether the Shanghai-based ADHM intervention can generalize to rural China or other LMICs without government-driven healthcare need further investigations” (page 10, line 4-7).

8. The role of financial incentives in physician behavior is not fully explored—Did incentives drive overprescription or unnecessary procedures?

Response: Thanks for pointing out this important issue. The financial incentives are not tied to prescriptions or referrals to cardiac procedures but rather to data entry on the

Shanghai AF management platform and regular follow-ups. Besides, the anticoagulation measures are based on CHA2DS2-VASc score and supervised by the administrative port of the platform. Therefore, the financial incentives did not affect the prescriptions or procedures on AF patients. This point is emphasized in the revised manuscript now (Supplementary Table 6, supplementary information page 16). It reads: “ Note that the financial incentives are not tied to drug prescriptions or referrals to cardiac procedures but rather to data entry on the Shanghai AF management platform and regular patient follow-ups.” .

9. The follow up term can be longer? 21 months is kind of in btw short and long terms, making it unclear whether benefits will persist.

Response: We followed the valuable suggestions. In the revised manuscript, we extended the follow-up term to 30 months, and the benefits persisted. Results were revised accordingly through the entire manuscript.

10. Another major problem is that the authors report lower costs in the ADHM cohort, but what is the cost-effectiveness ratio? The authors did not provide a breakdown of cost drivers nor consider indirect costs (e.g., patient travel, productivity loss). A cost expert needs to be involved to assess this with the more standard cost-assessment frameworks.

Response: Thanks for the insightful comment. The database did not contain those indirect costs. In the revised manuscript, we calculated the medical cost per patient per survival day (revised Table 2) and found that the medical costs were reduced by nearly 50 Chinese yuan per patient per survival day under ADHM model. The total cost were also presented in the Supplementary Table 4.

Medical costs per patient per survival day.

Cost per patient per survival day (Chinese Yuan*)	Total (n=8730)	ADHM cohort (n=1455)	UC cohort (n=7275)	P value
Total costs	255 ± 593	213 ± 266	263 ± 638	<0.001
Total costs in CHCs	53 ± 204	57 ± 177	52 ± 209	0.360
Total costs in hospitals [†]	202 ± 531	156 ± 191	211 ± 575	<0.001
Total outpatient costs	47 ± 93	60 ± 69	45 ± 98	<0.001
Outpatient costs in CHCs	22 ± 72	35 ± 61	20 ± 74	<0.001

Outpatient costs in hospitals	25 ± 57	25 ± 28	25 ± 62	0.998
Total inpatient/ED costs	207 ± 578	153 ± 255	218 ± 623	<0.001
Inpatient/ED costs in CHCs‡	31 ± 192	22 ± 167	33 ± 197	0.031
Inpatient/ED costs in hospitals	176 ± 519	131 ± 182	185 ± 562	<0.001

* Average exchange rate in 2023: 1 Chinese Yuan=0.142 US Dollars.

† Hospitals refers to higher-level hospitals.

‡ A part of CHCs have emergency and inpatient ward.

11. The KM curves in Figure 5 lack clear separation. Please consider adding time-dependent hazard ratios.

Reponse: The KM curves as well as the time stratified analyses were presented in the revised Figure 4, Supplementary Fig 2, and Supplementary Fig 3.

12. For Table 1, please highlight standardized mean differences (SMDs) to show balance across cohorts.

Response: Thanks and we followed the suggestion. The propensity score matching was performed and the SMDs were shown in the Supplementary Table 7.

13. Some of the long descriptive texts here and there (e.g., Shanghai AF Management Platform) should be shorten and moved to appendix.

Response: Thanks and we followed the suggestion. Related text is moved to Supplementary methods now.

Thanks again for your valuable inputs.

REVIEWER COMMENTS

Reviewer #1 (Remarks to the Author):

The authors have responded to all the comments satisfactorily.

Response: We thank the reviewer for the input and positive feedback.

Reviewer #2 (Remarks to the Author):

The response letter with specific answers was not uploaded.

Response: We uploaded the responses in the previous revision. Here we posted the responses below. Thank you very much for your input.

The present study is interesting, although not very representative for other countries around the globe. The presented concept is new. Nevertheless, questions remain:

The authors must provide data about the AF pattern (persistent, permanent or paroxysmal AF?) in the two groups. AF pattern has a substantial impact on outcome (see Goette et al JACC 2022).

Response: Thanks for the insightful comments. We agree the AF pattern is very important factors regarding cardiovascular outcomes. However, the current database adopted ICD-10 codes which did not differentiate persistent, permanent or paroxysmal AF (ICD-11 codes did, but not available in the current database). We have added the concern in the limitations (page 11, line 3-6). It reads: "Fourth, the pattern of AF, i.e., paroxysmal, persistent, or permanent AF, was not able to be differentiate due to the adoption of ICD-10 codes in the database rather than ICD-11 codes."

The overall rate of anticoagulation is rather low and not different in the two groups. Thus, how did the treatment protocol changes these treatment parameter to explain the observed differences in outcome?

Response: Thanks for pointing out this important issue. Yes, overall rate of anticoagulation was relatively low and similar between the two groups at baseline (approximately 40%). The rate was significantly increased after ADHM intervention (78.4%), compared with the UC cohort (46.3%), as shown in revised Figure 3A, 3B. Multivariate alaysis showed that regular anticoagulation was the independent protective factor of cardiovascular outcomes in the ADHM cohort (Supplementary Table 2). This point is clarified in the revised manuscript now.

Figure 3A,B Propotion of patients receicing anticoagulation.

Proportion of patients receiving at least one anticoagulant prescription (A), regular anticoagulation (B) at baseline and during follow-up.

What is the driving factor for the observed difference in outcome? How many patients had sinus rhythm at the end of the study in the two groups?

Response: Thanks for pointing out these important issues. These data are not available based on the data scan from ICD coding, and we described this study limitation in the revised manuscript now (page 11, line 6-7). It reads: “The status of rhythm (AF and its burden or sinus rhythm) at the end of the study, socioeconomic status and other health behaviors was also not available in the database.” Multivariate analyses were performed in the entire cohort, the ADHM cohort and the UC cohort, and the results were presented in the revised Supplementary Table 1, 2 and 3, respectively, showing the different management model (ADHM vs UC) was the independent driving factors of cardiovascular outcomes in patients with AF.

A multivariate analysis should be presented to determine the most significant effect on outcome for the two groups.

Response: Thanks and we followed the helpful comments. The results presented in the Supplementary Table 1, 2 and 3, and results section (page 7, line 3-10) now.

Supplementary Table 1. Predictors of the primary outcome in patients in both ADHM and UC groups.

Variables	Univariate		Multivariate	
	HR (95% CI)	P value	HR (95% CI)	P value
Baseline characteristics				

Age (years)	1.059 (1.052-1.065)	<0.001	1.053 (1.041-1.065)	<0.001
Female sex	1.286 (1.161-1.424)	<0.001	1.203 (0.972-1.490)	0.090
Hypertension	2.457 (1.894-3.188)	<0.001	1.176 (0.788-1.754)	0.427
Diabetes	1.381 (1.245-1.532)	<0.001	1.212 (0.974-1.509)	0.085
History of stroke/TIA	2.325 (2.088-2.588)	<0.001	1.779 (1.212-2.611)	0.003
History of SE	1.548 (1.128-1.885)	0.004	1.099 (0.839-1.440)	0.493
History of heart failure	1.793 (1.614-1.992)	<0.001	1.452 (1.163-1.812)	0.001
Periphery artery disease	1.690 (1.320-2.163)	<0.001	1.379 (1.065-1.787)	0.015
History of MI	1.458 (1.112-1.911)	0.006	1.392 (0.991-1.955)	0.057
Valvular heart disease	0.852 (0.647-1.120)	0.251		
COPD	1.289 (1.145-1.451)	<0.001	0.923 (0.813-1.048)	0.217
History of major bleeding	1.399 (1.202-1.629)	<0.001	0.940 (0.656-1.347)	0.738
History of cancer	0.953 (0.815-1.115)	0.549		
History of renal deficiency	1.971 (1.609-2.415)	<0.001	1.327 (0.899-1.959)	0.154
History of liver deficiency	1.556 (1.367-1.771)	<0.001	1.199 (0.843-1.705)	0.313
CHA2DS2-VASc score	1.355 (1.315-1.396)	<0.001	0.907 (0.752-1.093)	0.306
HAS-BLED score	1.658 (1.569-1.752)	<0.001	1.164 (0.842-1.609)	0.359
During follow-up				
UC compared to ADHM	1.674 (1.435-1.952)	<0.001	1.838 (1.556-2.173)	<0.001
Regular anticoagulant prescription	0.872 (0.788-0.966)	0.009	0.962 (0.862-1.073)	0.487
Antiarrhythmic drug prescription	0.998 (0.899-1.108)	0.966		
Rate control drug prescription	1.028 (0.921-1.148)	0.616		
Statin prescription	0.975 (0.838-1.135)	0.747		
RAASi prescription	1.024 (0.912-1.151)	0.685		
MRA prescription	1.075 (0.917-1.262)	0.372		
SGLT2i prescription	1.058 (0.832-1.346)	0.643		
Catheter ablation	1.173 (0.928-1.483)	0.181		
LAAC	1.347 (0.942-1.926)	0.102		
Visit CHCs≥4 times per year	0.912 (0.824-1.011)	0.078	1.246 (1.113-1.396)	<0.001
Times of visit to higher level hospitals	1.005 (0.984-1.026)	0.645		

Supplementary Table 2. Predictors of the primary outcome in patients in the ADHM group.

Variables	Univariate		Multivariate	
	HR (95% CI)	P value	HR (95% CI)	P value
Baseline characteristics				
Age (years)	1.099 (1.079-1.120)	<0.001	1.062 (1.028-1.097)	<0.001
Female sex	1.426 (1.069-1.902)	0.016	0.733 (0.379-1.419)	0.357
Hypertension	4.223 (1.703-10.469)	0.002	2.922 (0.777-10.985)	0.112
Diabetes	1.572 (1.167-2.117)	0.003	0.695 (0.351-1.378)	0.298
History of stroke/TIA	2.561 (1.877-3.494)	<0.001	1.454 (0.465-4.546)	0.520
History of SE	1.089 (0.508-2.332)	0.827		
History of heart failure	2.903 (2.111-3.992)	<0.001	1.010 (0.511-1.996)	0.978
Periphery artery disease	1.089 (0.545-2.178)	0.808		

History of MI	2.104 (1.097-3.034)	0.025	0.904 (0.351-2.329)	0.834
Valvular heart disease	1.395 (0.688-2.827)	0.356		
COPD	1.790 (1.305-2.453)	<0.001	1.102 (0.778-1.561)	0.585
History of major bleeding	1.712 (1.156-2.536)	0.007	3.199 (1.005-10.184)	0.049
History of cancer	1.325 (0.880-1.993)	0.177		
History of renal deficiency	1.740 (1.020-2.966)	0.042	3.353 (0.980-11.469)	0.054
History of liver deficiency	1.615 (1.131-2.307)	0.008	3.624 (1.122-11.701)	0.031
CHA2DS2-VASc score	1.533 (1.400-1.677)	<0.001	1.589 (0.887-2.847)	0.119
HAS-BLED score	1.752 (1.504-2.041)	<0.001	0.401 (0.134-1.200)	0.102
During follow-up				
Regular anticoagulant prescription	0.592 (0.429-0.816)	0.001	0.555 (0.384-0.801)	0.002
Antiarrhythmic drug prescription	1.012 (0.759-1.350)	0.935		
Rate control drug prescription	1.439 (1.070-1.936)	0.016	1.217 (0.874-1.695)	0.244
Statin prescription	1.515 (1.116-2.057)	0.747		
RAASi prescription	1.628 (1.219-1.173)	0.001	1.354 (0.984-1.864)	0.062
MRA prescription	2.465 (1.796-3.383)	<0.001	1.289 (0.899-1.848)	0.167
SGLT2i prescription	1.457 (0.937-2.265)	0.095	1.407 (0.848-2.334)	0.186
Catheter ablation	0.491 (0.288-0.837)	0.009	0.830 (0.468-1.470)	0.522
LAAC	1.403 (0.593-3.323)	0.441		
Visit CHCs≥4 times per year	0.576 (0.420-0.791)	0.001	0.971 (0.709-1.329)	0.855
Times of visit to higher level hospitals	0.951 (0.869-1.040)	0.269		

Supplementary Table 3. Predictors of the primary outcome in patients in the UC group.

Variables	Univariate		Multivariate	
	HR (95% CI)	P value	HR (95% CI)	P value
Baseline characteristics				
Age (years)	1.054 (1.047-1.160)	<0.001	1.046 (1.034-1.059)	<0.001
Female sex	1.269 (1.137-1.415)	<0.001	1.248 (0.994-1.566)	0.056
Hypertension	2.291 (1.744-3.011)	<0.001	1.016 (0.666-1.550)	0.940
Diabetes	1.359 (1.216-1.519)	<0.001	1.232 (0.976-1.555)	0.079
History of stroke/TIA	2.306 (2.056-2.587)	<0.001	1.634 (1.087-2.457)	0.018
History of SE	1.591 (1.208-2.097)	0.001		
History of heart failure	1.683 (1.505-1.883)	<0.001	1.010 (0.511-1.996)	0.978
Periphery artery disease	1.869 (1.429-2.444)	<0.001	1.454 (1.101-1.921)	0.008
History of MI	1.376 (1.022-1.853)	0.036	1.323 (0.916-1.911)	0.135
Valvular heart disease	1.148 (0.852-1.547)	0.363		
COPD	1.228 (1.080-1.396)	0.002	0.892 (0.778-1.022)	0.100
History of major bleeding	1.365 (1.157-1.610)	<0.001	0.783 (0.536-1.145)	0.208
History of cancer	1.106 (0.934-1.310)	0.244		
History of renal deficiency	2.051 (1.644-2.559)	<0.001	1.164 (0.770-1.759)	0.473
History of liver deficiency	1.550 (1.348-1.781)	<0.001	1.044 (0.721-1.512)	0.821
CHA2DS2-VASc score	1.335 (1.294-1.378)	<0.001	0.885 (0.725-1.079)	0.228
HAS-BLED score	1.652 (1.557-1.753)	<0.001	1.347 (0.959-1.893)	0.086

During follow-up				
Regular anticoagulant prescription	0.994 (0.891-1.110)	0.919		
Antiarrhythmic drug prescription	1.023 (0.914-1.145)	0.694		
Rate control drug prescription	1.085 (0.959-1.227)	0.195		
Statin prescription	0.964 (0.804-1.156)	0.692		
RAASi prescription	11.023 (0.897-1.167)	0.734		
MRA prescription	0.907 (0.749-1.099)	0.318		
SGLT2i prescription	1.064 (0.795-1.423)	0.678		
Catheter ablation	1.190 (0.288-0.837)	0.009	1.082 (0.801-1.462)	0.607
LAAC	1.830 (1.216-2.754)	0.004	1.914 (1.211-3.023)	0.005
Visit CHCs \geq 4 times per year	1.037 (0.930-1.157)	0.511		
Times of visit to higher level hospitals	0.989 (0.967-1.011)	0.320		

Reviewer #3 (Remarks to the Author):

Most of my comments are addressed in this revision. It is understandable that data limitation make some of my comments unaddressable.

Response: We thank the reviewer for the input and positive feedback.